

# Advancing Crop Modeling and Data Assimilation Using AquaCrop v7.2 in NASA's Land Information System Framework v7.5

Gabriëlle J.M. De Lannoy [1], Louise Busschaert [1], Michel Bechtold [1], Niccolò Lanfranco [1,2], Shannon de Roos [1,3], Zdenko Heyvaert [1,4], Jonas Mortelmans [1], Samuel A. Scherrer [1], Maxime Van den Bossche [1,5], Sujay Kumar [6], David M. Mocko [6,7], Eric Kemp [6], Lee Heng [8], Pasquale Steduto [9], and Dirk Raes [1]

[1]KU Leuven, Department of Earth and Environmental Sciences, Celestijnenlaan 200E, B-3001 Heverlee, Belgium
[2]Politecnico di Torino, Department of Environment, Land and Infrastructure Engineering, Corso Duca degli Abruzzi 24, 10129 Torino, Italy
[3]Vrije Universiteit Brussel, Department of Water and Climate, Pleinlaan 2, 1050, Brussel, Belgium
[4]ECMWF, Research Department, Reading, RG2 9AX, United Kingdom
[5]KU Leuven ICTS, Facilities for Research, HPC Support, B-3000 Leuven, Belgium
[6]NASA/GSFC, Hydrological Sciences Laboratory, Greenbelt, MD 20771, USA
[7]Science Applications International Corporation, Reston, VA, USA
[8]Formerly* at International Atomic Energy Agency, Vienna, Austria
[9]Formerly* at Land and Water Division, FAO, Rome, Italy

**Correspondence:** Gabriëlle J.M. De Lannoy (gabrielle.delannoy@kuleuven.be)

**Abstract.** This paper introduces the open-source AquaCrop v7.2 model as a new process-based crop model within NASA's Land Information System Framework (LISF) v7.5. The LISF enables high-performance crop modeling with efficient geospatial data handling, and paves the way for scalable satellite data assimilation into AquaCrop. Through three exploratory showcases, we demonstrate the current capabilities of AquaCrop in the LISF, along with topics for future development. First, coarse-scale

crop growth simulations with various crop parameterizations are performed over Europe. Satellite-based estimates of land surface phenology are used to inform spatially variable crop parameters. These parameters improve canopy cover simulations in growing degree days compared to using uniform crop parameters in calendar days. Second, ensembles of coarse-scale simulations over Europe are created by perturbing meteorological forcings and soil moisture. The resulting uncertainties in root-zone soil moisture and biomass are often greater in water-limited regions than elsewhere. The third showcase aims to improve fine-

scale agricultural simulations through satellite data assimilation. Fine-scale canopy cover observations are assimilated with an ensemble Kalman filter to update the crop state over winter wheat fields in the Piedmont region of Italy. The state updating is beneficial for the intermediary biomass estimates, but leads to only small improvements in yield estimates relative to reference data. This is due to strong model (parameter) constraints and limitations in the assimilated satellite observations and reference yield data. The showcases highlight pathways to improve or advance future crop estimates, e.g. through crop parameter

updating and multi-sensor and multi-variate data assimilation.

---

*retired





# 1 Introduction

Food production, agricultural land and water management, and their interaction with socio-economic demands are facing increasing challenges in our changing world. Tapping into advanced computational models and an abundance of satellite data on crop-water systems, we can enhance our ability to understand, monitor, and manage these challenges from the level of single fields all the way to the global scale.

Crop models allow us to dynamically simulate crop phenology and derive yields and irrigation needs, given meteorological and soil input along with crop information. These models can be grouped into (i) biophysical mechanistic or process-based crop models, and (ii) empirical or statistical crop yield models, including machine learning models. Although the latter data-driven models are rapidly gaining attention and precision (van Klompenburg et al., 2020; Paudel et al., 2022; Gaso et al., 2024), the former remain indispensable as a reference to understand interactions between crop phenology and the environment, and to estimate poorly observed variables such as root-zone soil moisture. Most process-based crop models were originally developed for field (<1 ha) applications, and are now increasingly used for regional to global applications. This is stimulated by the growing availability of satellite-based data to serve as input, to calibrate parameters, or to update the model during simulation via data assimilation (DA) for larger regions. Advancing large-scale satellite-based crop DA requires computationally efficient and scalable frameworks. This includes flexible ways (i) to drive crop models with various types of parameters and meteorological forcings, (ii) to produce uncertainty estimates via ensemble simulations, and (iii) to interface models with various types of satellite data using a range of DA techniques. The NASA Land Information System Framework (LISF, Kumar et al. (2006, 2008)) offers such a framework, but so far it was limited to hosting land surface and hydrology models. Some of these models allow simulations of crop growth and agricultural practices, such as irrigation (Busschaert et al., 2025). However, land surface models are designed to ultimately estimate water, energy and carbon interactions between the land and the atmosphere, and thus typically serve to e.g., initialize weather forecasts, and hydrology models aim at simulating the water budget, floods and droughts. By contrast, crop models aim at simulating crop yield, and the yield response to water availability, with more land management options, a more detailed description of crop phenology stages, and a refined stage-specific sensitivity to various stresses. This paper builds on the long-standing development of LISF to explore the potential of crop DA, by integrating a crop model into the LISF. Specifically, AquaCrop v7.2 is integrated within LISF v7.5, and various capabilities of this new system are demonstrated through three exploratory showcases.

In the realm of process-based crop models, AquaCrop (Raes et al., 2009; Steduto et al., 2009) is a water-driven crop model with relatively limited input requirements. Launched in 2009 to simulate the yield response of herbaceous crops to water availability at the field scale, the model now includes, for example, the simulation of perennial herbaceous forage crops (e.g. alfalfa), salinity and improved biomass-stress relationships. AquaCrop comes with a database of calibrated crop parameters for a range of crops, but specific crop calibration is often recommended (Wellens et al., 2022). The sensitivity of yield simulations to crop (and soil) parameters has been well studied for AquaCrop at the field scale, both in humid and dry regions (Vanuytrecht et al., 2014; Lu et al., 2021). More recently, AquaCrop has been used for coarse-scale regional to global simulations of soil moisture and biomass (de Roos et al., 2021), irrigation (Busschaert et al., 2025), water footprints (Mialyk et al., 2024), food-energy-



water nexus analyses (Akbari Variani et al., 2023), and to quantify the impact of future climate on irrigation requirements and yield (Busschaert et al., 2022, in review). While many of these applications are interested in the yield of specific crops within a coarse-scale grid cell, other coarse-scale applications require generic crop parameters to represent the patchwork of fields within a grid cell. In a first showcase using AquaCrop in the LISF, we will demonstrate that such generic crop parameters for coarse-scale AquaCrop simulations can be derived from the satellite-based Global Land Surface Phenology product (GLSP;

Zhang et al., 2018) over Europe.

The performance of the crop model depends not only on parameter choices but also on other input, and the model structure. Multi-model ensembles have been used to quantify the modeled yield response to temperature, $CO_2$, fertility and management, for within-season agricultural forecasting and climate simulations (Müller et al., 2021). For short-term to seasonal yield or irrigation forecasts with a single model, weather input in particular strongly influences the quality of simulations (Challinor

et al., 2005; Busschaert et al., 2024; Zare et al., 2024). The initial crop and soil moisture state are also crucial in determining the trajectory of crop and water balance forecasts. By perturbing model input, parameters, and/or the state, an ensemble of forecasts can be created to quantify the forecast uncertainty or to analyze the sensitivity of particular crop variables to input uncertainty. This is useful in itself, or to facilitate DA in a next step. In a second showcase of this paper, we will illustrate how the uncertainty in coarse-scale AquaCrop biomass simulations is related to uncertainty in soil moisture and weather conditions

over Europe.

In the context of crop modeling, DA aims to optimally integrate observed data to improve crop estimates through parameter, forcing, or state estimation (Evensen et al., 2022). DA also adds value to observations by inferring unobserved variables of the crop system, e.g. by estimating root-zone soil moisture from surface soil moisture or vegetation observations. Field-scale crop DA has been explored using in-field, satellite, or synthetic observations, both for parameter and state estimation (Pauwels

et al., 2007; Linker and Ioslovich, 2017; Lu et al., 2021, 2022; Zare et al., 2024; Yang and Lei, 2024). For larger domains, crop DA has mainly used satellite observations of soil wetness and crop biophysical variables (Jin et al., 2018) to estimate soil or crop parameters (Dente et al., 2008; Huang et al., 2023). However, regional to global sequential satellite DA to improve crop yield forecasts via soil moisture or crop state updating (de Wit and van Diepen, 2007; de Roos et al., 2024) is still in its infancy compared to current operational or state-of-the-art assimilation systems for land surface models (Reichle et al., 2019;

De Lannoy et al., 2022). In a third showcase, we will explore the assimilation of fine-scale satellite observations of fractional vegetation cover (FCOVER) to refine biomass and yield estimates for the Piedmont area in Italy, Europe.

The objective of this paper is to introduce the open-source AquaCrop v7.2 model as a dynamical state propagation model within NASA's LISF v7.5 (Section 2), and to pave the way for large-scale satellite DA into a crop model. The latter is achieved in three showcases, demonstrating the current potential and limitations of (i) coarse-scale crop growth simulation with various

crop parameterizations, (ii) coarse-scale ensemble simulations, and (iii) ensemble Kalman filtering of fine-scale satellite data into AquaCrop. The background and methods for the three showcases are given in Section 3, the results and discussion in Section 4, and the paper ends with conclusions in Section 5.



## 2 Modeling and Data Assimilation Framework

### 2.1 AquaCrop v7.2 Assets

Originally, AquaCrop of the Food and Agriculture Organization (FAO) was developed in a Delphi/Pascal programming language to be used in a graphical user interface (GUI). In 2022, the core engine of AquaCrop version 7.0 (v7.0) was converted to Fortran 2003 and optimized, to promote wider use, transparent version control, and long-term community-based maintenance on GitHub. This conversion was done in a semi-automatic way by KU Leuven researchers with guidance from KU Leuven high-performance computing (HPC) support, and extensively verified via a range of testcases covering all of Europe and the full range of crop and management scenarios. Fortran is a green programming language, widely used in the earth and climate sciences community, and allows easy interfaces with other languages. Unlike older open-source versions in other languages (Kelly and Foster, 2021), the Fortran code includes all AquaCrop functionalities (e.g. salinity, fertility, perennial crops, etc.). At the time of writing, AquaCrop v7.2 is the latest released version, distributed through https://github.com/KUL-RSDA/AquaCrop (GitHub) and documented on https://ees.kuleuven.be/en/aquacrop.

The five key AquaCrop assets are summarized in Figure 1. They include (i) the GUI application, (ii) the open-source version-controlled Fortran code available on GitHub, (iii) the stand-alone programs for Windows, MacOS, and Linux, compiled from the Fortran code, (iv) a simple Python wrapper around the stand-alone program to run AquaCrop in parallel across grid cells (https://github.com/KUL-RSDA/RegionalAC_Py), and (v) the integration of the Fortran code into NASA's LISF (https://github.com/NASA-LIS/LISF, Section 2.3). The first three assets are made available with each version release on GitHub (https://github.com/KUL-RSDA/AquaCrop/releases) and offer three forms of the AquaCrop core engine. The GUI comes with an FAO copyright. The last two assets employ the stand-alone executable and plain Fortran code, respectively, to allow efficient regional- to global-scale modeling, climate impact simulations, and satellite DA (de Roos et al., 2021; Busschaert et al., 2022, 2025, in review; de Roos et al., 2024).

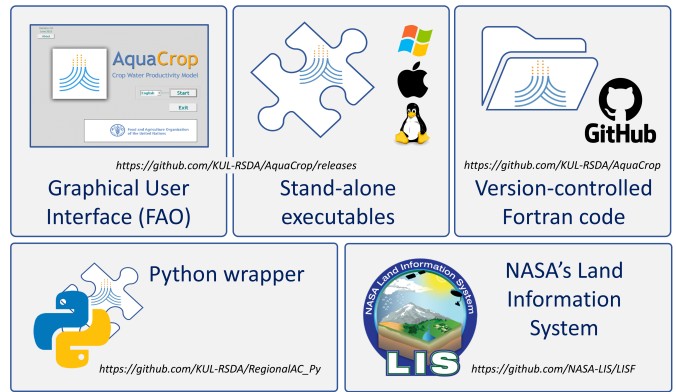

**Figure 1.** The five main AquaCrop v7.2 assets: three forms of the AquaCrop core engine (top) and two derived assets to support regional simulations (bottom).





## 2.2 AquaCrop v7.2 Processes

AquaCrop computes, for each daily timestep $i$, the (1) canopy development, (2) water balance, (3) biomass, and (4) yield, as shown in the flowchart of Figure 2. We summarize the model here with a focus on the crop and soil water state propagation in time, and with special attention to the system's time-variant prognostic and diagnostic variables (with subscript $i$), and time-invariant parameters (without subscript $i$). Prognostic or state variables depend on state estimates of a previous time step, whereas diagnostic variables are derived from the current state variables. Like most other crop models, AquaCrop can use either calendar time or thermal time (also known as growing degree days, GDD) to track crop development.

For each day, the prognostic canopy cover $CC_i$ [m$^2$.m$^{-2}$] describes the crop phenology development, either based on calendar days or GDD [°C day]. First, we focus on the potential crop canopy cover evolution, $CC_{\mathrm{pot},i}$ [m$^2$.m$^{-2}$] without crop stress. The gray area in the first part of Figure 2 reflects $CC_{\mathrm{pot},i}$ and is determined by a stage-dependent piecewise function $f_{\mathrm{cc}}(i, \boldsymbol{\alpha}_{\mathrm{cc}})$, described in Appendix A. This function depends on time $i$ and several parameters, collectively referred to as the parameter vector $\boldsymbol{\alpha}_{\mathrm{cc}} = [CC_{\mathrm{o}}, CC_{\mathrm{x}}, CGC, CDC, \ldots]$. $CC_{\mathrm{o}}$ [m$^2$.m$^{-2}$] is the initial $CC_{\mathrm{pot},i}$ at emergence. Thereafter, the growth sets in to reach $CC_{\mathrm{x}}$ [m$^2$.m$^{-2}$] or maximal $CC_{\mathrm{pot},i}$, with growth defined by the canopy growth coefficient $CGC$ in [day$^{-1}$] or [°C.day$^{-1}$]. At the end of the season, the canopy decline coefficient $CDC$ in [day$^{-1}$] or [°C.day$^{-1}$] describes the rate of green canopy decline to the final stage of crop maturity, which marks the end of the crop growth season. Again, $CC_{\mathrm{pot},i}$ solely depends on time $i$ and $\boldsymbol{\alpha}_{\mathrm{cc}}$, and serves as a parameterized upper limit to the actual $CC_i$. By contrast, the actual $CC_i$ computation accounts for soil water, soil fertility and soil salinity stresses, by substituting $\boldsymbol{\alpha}_{\mathrm{cc}}$ with rescaled equivalents $\boldsymbol{\alpha}_{\mathrm{cc},i}$, i.e. the parameters are dynamically adjusted for $CC_{i-1}$ and water, fertility and salinity stresses:

$$CC_{\mathrm{pot},i} = f_{\mathrm{cc}}(i, \boldsymbol{\alpha}_{\mathrm{cc}}) \tag{1}$$
$$CC_i = f_{\mathrm{cc}}(i, \boldsymbol{\alpha}_{\mathrm{cc},i}(CC_{i-1}, \boldsymbol{\theta}_i)) \tag{2}$$

where $\boldsymbol{\theta}_i$ here refers to the vector of prognostic soil water estimates in all soil compartments for simplicity, but to be more precise, it also includes salinity and soil fertility. In the equations of this paper, we specifically choose to highlight the dependencies on the soil water ($\in \boldsymbol{\theta}_i$) and $CC_i$ state variables, and we refer to Raes et al. (2025b) for a full model description.

The piecewise $f_{\mathrm{cc}}(.)$ is thus a set of different exponential growth and decay functions (Steduto et al., 2009; Raes et al., 2009), constrained by time (or GDD) and crop parameters $\boldsymbol{\alpha}_{\mathrm{cc}}$ (incl. crop stages), in such a way that $CC_i \leq CC_{\mathrm{pot},i}$. This means that for a model trajectory with a given parameter $CC_{\mathrm{x}}$, the $CC_i$ should or could never be perturbed or updated above the corresponding $CC_{\mathrm{pot},i}$. Furthermore, if soil fertility stress is set, the $CC_{\mathrm{x}}$ will be modulated by the fertility budget to $CC_{\mathrm{x},\mathrm{sf},i}$ assuming no water stress, and the attainable potential canopy cover is $CC_{\mathrm{pot},\mathrm{sf},i}$, i.e. $CC_i \leq CC_{\mathrm{pot},\mathrm{sf},i} \leq CC_{\mathrm{pot},i}$ (see Section 3.3, and Appendix A). Soil fertility lumps the effect of field management (pest control, nutrients, post-harvest loss, . . . ) on crop production.

After rescaling for micro-advective affects, the variable $CC_i^*$ is used in the second step to diagnose transpiration $Tr_i$ [mm.day$^{-1}$]:

$$Tr_i = K_{\mathrm{s},i}(\boldsymbol{\theta}_i) \times [CC_i^*(\boldsymbol{\theta}_i) \times K_{c,\mathrm{Tr},\mathrm{x},i}] \times ET_{\mathrm{o},i} \tag{3}$$





with $ET_{\mathrm{o},i}$ [mm.day$^{-1}$] the input reference grass evapotranspiration, $K_{\mathrm{s},i}$ [-] ($0 \leq K_{\mathrm{s},i} \leq 1$, with 1 being no stress) a stress coefficient accounting for cold and soil water stress (diagnosed from $\boldsymbol{\theta}_i$, i.e. water shortage and logging, and also salinity), and $K_{c,\mathrm{Tr,x},i}$ [-] is the maximum crop transpiration coefficient that depends on the specific crop, and is adjusted for aging, senescence and elevated $CO_2$. After normalizing the $Tr_i$ for the climatic conditions using $ET_{\mathrm{o},i}$ [mm.day$^{-1}$], it is used to update the prognostic dry aboveground biomass production $B_i$ [t.ha$^{-1}$] via a crop-specific water productivity factor $WP_i^*$ [t.ha$^{-1}$.day$^{-1}$], which is normalized for the effect of climatic conditions (meteorology and $CO_2$ concentration), adjusted for the crop stage, and rescaled by a soil fertility stress coefficient $K_{\mathrm{sf},i}$ [-] ($0 \leq K_{\mathrm{sf},i} \leq 1$, with 1 being no stress):

$$
\begin{aligned}
B_i &= B_{i-1} + \Delta B_i \Delta t & (4) \\
\Delta B_i &= K_{\mathrm{sf},i} \times WP_i^* \times \frac{Tr_i(\boldsymbol{\theta}_i, CC_i)}{ET_{\mathrm{o},i}} & (5)
\end{aligned}
$$

$\Delta B_i$ is thus the biomass production per model time step [t.ha$^{-1}$.day$^{-1}$], and $\Delta t$ refers to the daily model time step. Finally, dry yield $Y_i$ [t.ha$^{-1}$] is diagnosed from biomass by multiplication with a crop-specific harvest index $HI_{\mathrm{o}}$, [-] and an adjustment factor $K_{\mathrm{HI},i}$ [-] that accounts for time, a crop-specific growth coefficient, and water, heat and cold stresses at different stages in the growing cycle:

$$
Y_i = K_{\mathrm{HI},i}(\boldsymbol{\theta}_i) \times HI_{\mathrm{o}} \times B_i(\boldsymbol{\theta}_i, CC_i) \tag{6}
$$

The harvested $Y_i$ is obtained at the time of crop maturity.

The equations above are deliberately written in a way that highlights their dependence on $\boldsymbol{\theta}_i$. The prognostic soil water, salinity and fertility budgets are computed at each time step. These three budgets are driven by meteorological input of daily temperature, precipitation (rainfall) and $ET_{\mathrm{o}}$, and in turn these budgets drive the stresses in AquaCrop. Here, we focus only on the soil water budget. A soil reservoir with 12 compartments (prognostic state variables $\in \boldsymbol{\theta}_i$) receives water through infiltration of precipitation and possibly irrigation $P_i$, draws water from shallow groundwater through capillary rise $C_i$, and releases water through deep percolation $D_i$, soil evaporation $E_i$ and crop transpiration $Tr_i$. These water fluxes [mm.day$^{-1}$] are cumulated over 1 day (from $i-1$ to $i$) and move through the profile following soil physical laws represented by functions $f_{\boldsymbol{\theta}}(.)$, that result in a change in soil moisture $\Delta \boldsymbol{\theta}_i$, as follows:

$$
\begin{aligned}
\boldsymbol{\theta}_i &= \boldsymbol{\theta}_{i-1} + \Delta \boldsymbol{\theta}_i \Delta t & (7) \\
\Delta \boldsymbol{\theta}_i &= f_{\boldsymbol{\theta}}(P_i, C_i, D_i, E_i, Tr_i) & (8)
\end{aligned}
$$

The water stored in the dynamic root zone determines some of the crop growth stresses mentioned above. The prognostic root-zone depth $Z_i$ [m] is a parameterized function of time (calendar days or GDD), modulated by soil water availability as follows:

$$
\begin{aligned}
Z_i &= Z_{i-1} + \Delta Z_i \Delta t & (9) \\
\Delta Z_i &= K_{\mathrm{z},i}(\boldsymbol{\theta}_i) \left[ f_{\mathrm{z}}(i, \boldsymbol{\alpha}_{\mathrm{z}}) - f_{\mathrm{z}}(i-1, \boldsymbol{\alpha}_{\mathrm{z}}) \right] & (10)
\end{aligned}
$$

where $f_{\mathrm{z}}(.)$ is a set of functions that describe the potential rooting depth solely as function of time (or GDD) and time-invariant parameters $\boldsymbol{\alpha}_{\mathrm{z}}$, containing the minimal and maximal potential rooting depth parameter, illustrated in Figure 3. $K_{\mathrm{z},i}(\boldsymbol{\theta}_i)$ [-]





reduces the maximal potential expansion of the rooting depth, based on a diagnosis of stomatal stress (determined by root-zone soil moisture) or dry subsoil at the front of the root-zone expansion. Just like $CC_i$ is double bounded by $f_{cc}(.)$, so $Z_i$ is double bounded by $f_z(.)$ as a strong model constraint determined by parameters.

Eqs. 2, 3, 6 and 10 in particular contain piecewise functions and thus discontinuities that are determined by crop stages, i.e., the crop-water system propagates through different crop development regimes. AquaCrop considers the following phenological stages, which are parameterized in either calendar days or GDDs, and marked on Figure 3: (I) time to emergence, i.e. the onset of greening, (II) time to maximum rooting depth, (III) time to flowering, (IV) time to senescence, and (V) time to maturity, i.e. the end of the crop cycle. A particularity of AquaCrop (and many other crop models) is that the timing of the crop stages is assumed to be known at the beginning of the crop simulation, i.e. it is parameterized. The entire temperature record of a simulation year is used to precompute the timing of the stages at the beginning of that simulation year. This limitation will be addressed in future model development and results from the original purpose of AquaCrop, which was to use historical meteorological data to test strategies to improve sustainable crop production in the future.

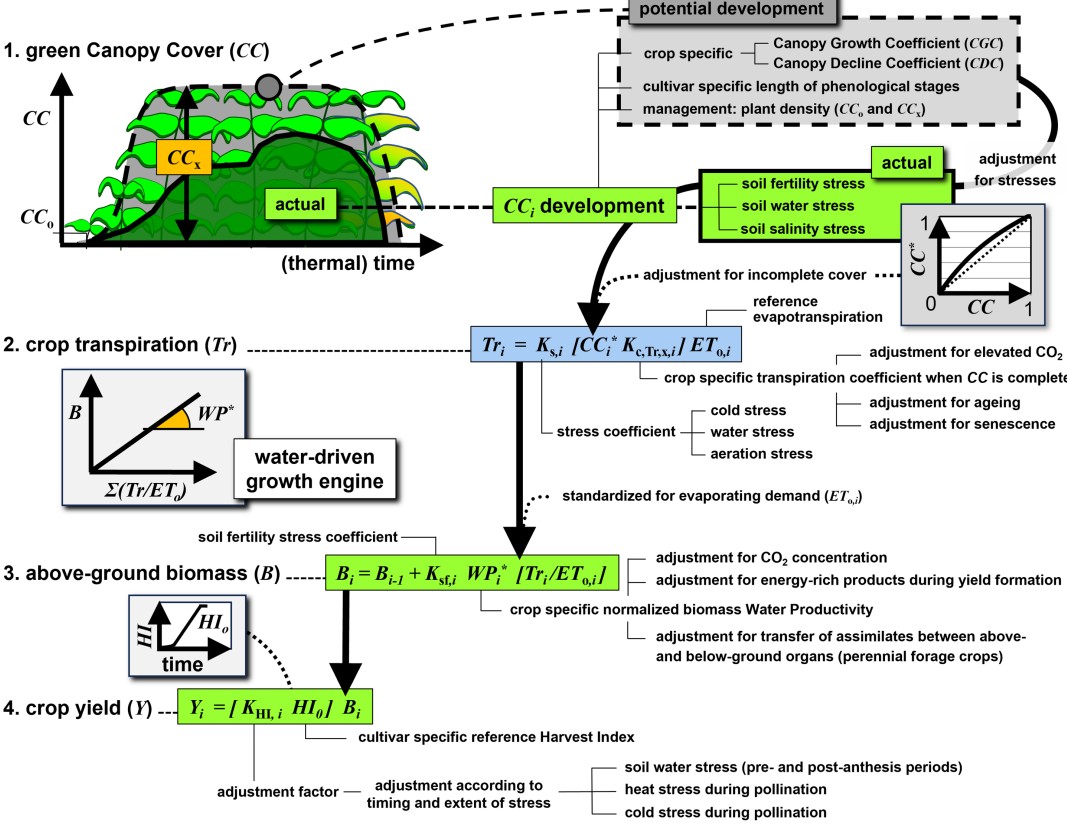

**Figure 2.** Schematic diagram of the AquaCrop model with the four main computational steps from crop development to yield, adapted from Raes et al. (2025a). Symbols are explained in the text.



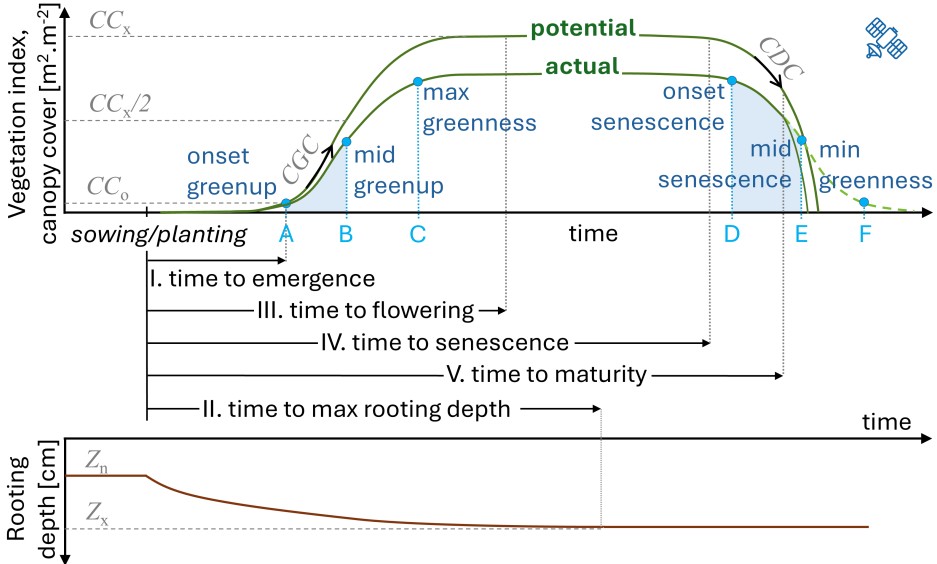

**Figure 3.** Crop stages assumed in (black I-V) AquaCrop, with an indication of some other (gray) essential AquaCrop crop parameters that define the $CC_{\mathrm{pot}}$ and rooting depth evolution. The timing of the crop stages can be inferred from (blue A-F) the time of the various satellite-based phenology stages in the GLSP product. Note the difference in the assumed shape of the vegetation curve for (dark green) AquaCrop and (dashed light green) GLSP during senescence.

## 2.3 NASA LISF v7.5

NASA's LISF is a scalable software framework with three components: (i) the Land surface Data Toolkit (LDT) to handle the data-related requirements to prepare for model simulations, (ii) the Land Information System (LIS) that integrates multiple land surface models (LSMs), satellite data readers, and DA techniques (Kumar et al., 2006, 2008), and (iii) a Land surface
Verification Toolkit, not yet used in this paper. It provides a portable infrastructure to transition Earth science research into operations (Kumar et al., 2019), can be coupled to Earth system models (Heyvaert et al., 2025), and it offers a digital twin environment for land surface hydrology (De Lannoy et al., 2024) and from now on also for agricultural cropping systems. LISF utilizes high-performance computing along with data management technologies to address the computational difficulties associated with high-resolution land surface modeling.
So far, LIS has hosted a range of LSMs, but no dedicated crop models yet. AquaCrop v7.2 is coupled into LIS v7.5 using shared plugins and standard interfaces. Because the model was not originally designed to serve as a state propagation model within an assimilation system, and because it behaves differently than the other models already embedded in LIS, some technical aspects required attention. First, AquaCrop cannot be run at subdaily resolution, whereas most LSMs are typically run at subhourly resolutions. This means that meteorological forcings, typically provided at an hourly resolution, must be aggregated
to daily values, and the maximum and minimum air temperatures of the day must be extracted to be used directly as input to the crop model and in the calculation of $ET_{\mathrm{o}}$. Second, AquaCrop relies on numerous global variables, which include state



variables (e.g. soil moisture, biomass) as for the LSMs, but also many other variables that are diagnosed from or not at all related to state variables, and are necessary to describe the current crop system conditions. These include flags that are turned on and off to mark a certain process (e.g. early senescence). These global variables are passed on from one timestep (day) to another, and therefore need to be saved to restart the simulation later on. Third, the time-stepping mechanism and model progression in AquaCrop were originally integrated into AquaCrop's own file management system. To enable coupling with the LIS, a new routine is developed that enables the model to advance by a single time step when called externally, following after the initialization of all required global variables.

AquaCrop requires input on meteorology and $CO_2$ (collectively called 'climate input' in AquaCrop literature), soil, crop, and management. In LIS, the meteorology is typically read from global netcdf files of re-analysis products. The atmospheric $CO_2$ concentrations are taken from measurements at the Mauna Loa Observatory in Hawaii (Thoning et al., 2025). Spatially distributed soil and crop type classes are entered via geospatial netcdf files that are created via the LDT as front-end processor for LIS, and then mapped to the associated soil hydraulic or crop parameters via an AquaCrop lookup table and crop parameter file, respectively. The latter is provided in the configuration file of LIS. Likewise, management options, such as those related to fertility and irrigation, are provided in the LIS configuration file via an AquaCrop parameter file.

Currently, the configuration file for LIS is limited to enabling just a few of the many available AquaCrop options. Inactive features that might be introduced in the future include soil profiles with varying textures at different depths, varying groundwater levels, crop rotation, perennial crops, and salinity stress. Furthermore, only a limited set of model parameters are currently spatially variable. Most of the management parameters (soil fertility, irrigation) are uniform, but this can easily be addressed in future versions.

Through three exploratory showcases (Section 3) at different spatial scales, we demonstrate some AquaCrop applications that are facilitated by LIS. First, LIS enables us to efficiently handle spatially distributed input forcings and parameters at any resolution. In section 3.1, we constrain coarse-scale spatially distributed generic crop parameters with satellite data, and then use them in AquaCrop simulations over Europe. Second, LIS facilitates ensemble perturbations of the forcings, parameters and state variables, possibly with spatial and temporal correlations, and ensemble perturbation bias correction. In Section 3.2, forcings and soil moisture state variables are perturbed to demonstrate their impact on biomass simulations for Europe. Finally, LIS hosts different DA algorithms and interfaces with a range of satellite data. In Section 3.3, the potential and current limitations of high-resolution FCOVER satellite DA are illustrated for the Piedmont region of Italy. Note that the time index $i$ is omitted from the time-varying variables in the following for simplicity, unless needed for clarity (in Section 3.3 only).



## 3 Methods

### 3.1 Showcase 1: Crop Parameterization

#### 3.1.1 Background

AquaCrop was originally designed to simulate a generic annual herbaceous crop. With time, crop parameters were adjusted via calibration to simulate various specific herbaceous species, and perennial forage crops were included. The AquaCrop crop parameters consist of so-called conservative parameters that are independent of climate, time, location, management, or cultivar, and other parameters that depend on, e.g., the cultivar or planting mode and should be calibrated. The most important crop parameters are related to the length of the different stages of phenological development, $CC_x$, $GCG$ and $CDC$, marked in Figure 3, and $HI_o$.

Specific crop parameters (e.g. for maize, wheat, ...) are necessary for applications that aim at crop-specific yield estimates (Busschaert et al., in review) or parcel management. Such parameters can be derived from field data or high-resolution (10-100 m) satellite data (Franch et al., 2022; Gobin et al., 2023). Including satellite-based specific crop phenology information has proven to improve crop model simulations (Bregaglio et al., 2023). In contrast, generic crop parameterizations are needed for coarse-scale simulations of crop growth, irrigation, and overall carbon and water budgets in mixed crop grid cells (de Roos et al., 2021; Busschaert et al., 2022), and cannot be used to estimate specific crop yield. Generic crop parameters for mosaics of crops with different crop stages in rotating field locations are hard to define, but effective spatial climatological patterns of phenological metrics can be derived from lower-resolution (>100 m) satellite data (Hmimina et al., 2013; Zhang et al., 2018). To our knowledge, no study has reported whether satellite-based parameterization of a generic crop can improve coarse-scale AquaCrop simulations.

#### 3.1.2 Experiment

We evaluate coarse-scale AquaCrop simulations with two generic crop configurations, with parameters summarized in Appendix C. The first crop configuration defines the development stages in spatially uniform calendar days, as in de Roos et al. (2021) for C3 crops. It is solely based on agronomist expertise and tailored to obtain good coarse-scale biomass estimates. The second approach uses spatially variable parameters in GDD mode. These parameters are derived from maps of day of the year (DOY) for various crop stages provided by the 0.05° Global Land Surface Phenology product of the Visible Infrared Imaging Radiometer Suite (VIIRS GLSP; Zhang et al., 2018) for the period 2013-2022 (10 years). The mapping between GLSP stages and AquaCrop parameters is summarized in Table 1, illustrated in Figure 3 and details are provided in Appendix A. In short, for each year, the first growing season is extracted from the the GLSP product, the DOY for each particular stage in each 0.05° grid cell is converted into a GDD value using re-analysis temperature, and the $CGC$ and $CDC$ parameters are estimated by inverting the $f_{cc}(.)$ functions (Eq. 2, Appendix A). Next, the median 0.05° crop parameters are computed across the 10 years.

Each of both generic crop parameterizations is used in an AquaCrop simulation experiment for the period 2015 through 2020. The meteorology is taken from the Modern-Era Retrospective Analysis version 2 (MERRA-2; Gelaro et al., 2017) and



**Table 1.** Link between AquaCrop parameters and VIIRS-GLSP crop stages, marked in Figure 3, and explained in Appendix A. For AquaCrop experiments in GDD mode, the times for the crop stages (A, D, F) are first converted from DOY to GDD before deriving the time to maximal rooting depth, $CGC$ or $CDC$.

| AquaCrop | VIIRS-GLSP |
|---|---|
| I. time to emergence | A |
| II. time to max rooting depth | 0.7·D *[estimated]* |
| III. time to flowering | N/A *[only needed for specific determinate crops]* |
| IV. time to senescence | D |
| V. time to maturity | F |
| $CGC$ | $\frac{\ln \frac{CC_\mathrm{x}/2}{CC_\mathrm{o}}}{\Delta t_\mathrm{A \to B}}$ |
| $CDC$ | $\frac{2.40(CC_\mathrm{x}+2.29)}{3.33\Delta t_\mathrm{D \to E}}$ |

bilinearly interpolated to the model grid. The simulations are performed at $0.1°$ resolution for all grid cells with dominant crop-land in Europe according to the 1-km global land cover data set from the University of Maryland (Hansen et al., 1998). The satellite-constrained parameters in GDD are aggregated from the $0.05°$ to the $0.1°$ resolution, taking the average of all $0.05°$

grid cells with dominant cropland. The soil texture is taken from the Harmonized World Soil Database 1.21 (FAO/IIASA/IS-RIC/ISSCAS/JRC, 2012) as a weighted combination of surface and subsurface texture, following De Lannoy et al. (2014), and the texture is mapped to default soil hydraulic parameters for AquaCrop v7.2 (Raes et al., 2025b). To parameterize field management, a uniform soil fertility stress of 30% is assumed for both experiments as in de Roos et al. (2021). A deterministic run for one simulation year and 33,670 crop grid cells takes about 45 min, when run on 36 central processing units (cpus). The

compute times reported in this paper are conservative and measured on the Tier-1 Hortense and Tier-2 wICE clusters of the Vlaams Supercomputer Centrum (VSC).

The daily results of simulated $CC$ and $\Delta B$ are averaged to 10-day values and evaluated against independent satellite data, i.e. Copernicus Global Land Service (CGLS) Fraction of Vegetation Cover (FCOVER) and Dry Matter Productivity (DMP) for both simulations. The performance is quantified for the period between the first and last timestep for which any of the two

AquaCrop simulations or the CGLS DMP reference data exceed 5% of their maximum $\Delta B$ value in the year. This period is referred to as the 'maximal growing season' below.

## 3.2 Showcase 2: Ensemble Simulations

### 3.2.1 Background

Crop model simulations are never perfect, due to uncertainties in crop, soil and management parameters, model structure,

meteorological input, and initial conditions. Sampling a range of possibilities for these aspects allows to create an ensemble of crop model trajectories and to quantify (i) the time-varying sensitivity of various simulated crop variables to these aspects, (ii) the uncertainty of the simulations, and (iii) the correlation of the forecast errors between the various simulated variables.



Dynamic ensemble uncertainty estimates are particularly important for DA in a next step. With the exception of a few studies geared towards DA for state updating (de Roos et al., 2024; Lu et al., 2022), most ensemble simulations with AquaCrop have been performed to study the model sensitivity to crop parameters, and not to errors in the state or meteorological estimates.

### 3.2.2 Experiment

Because AquaCrop is a water-driven crop model, $CC$ and $B$ simulations depend on estimates of the root-zone soil moisture. To quantify this dependency, we utilize the same setup as in showcase 1 above for the generic crop with GDD parameterization over Europe, but we now perturb MERRA-2 meteorological forcings and the soil moisture state variables in the top 10 compartments to generate an ensemble of 24 members. This uncertainty will propagate to uncertainty in $CC$ and $B$. The perturbations parameters are spatially and temporally constant as summarized in Table 2. The resulting perturbations are applied hourly to shortwave radiation and precipitation, and daily to soil moisture estimates. The hourly perturbed MERRA-2 data are converted to daily AquaCrop forcing input of $ET_\mathrm{o}$ and $P$ as in Busschaert et al. (2022). Note that unlike de Roos et al. (2024) (who used a calendar day crop parameterization), temperature is not perturbed, because this can create inconsistencies with the precomputed crop growth stages in GDD (see Section 2.2). A perturbation bias correction (Ruy et al., 2008) is applied to keep the soil moisture ensembles centered around the unperturbed deterministic simulation. Because the structure and parameters of the model are kept fixed, the model here acts as a strong constraint. An ensemble run for 33,670 crop grid cells with 24 members for one simulation year takes about 90 min, when run on 128 cpus.

Simulations for 3 years, from 2015 through 2018, are used to compute the multi-year average ensemble standard deviation (also called 'spread' below) in root-zone soil moisture and $B$ for all croplands in Europe. For simplicity, the soil moisture spread for the maximal rooting depth (0-100 cm, 10 compartments) is computed and averaged over 3 years without accounting for the varying rooting depth and without masking for the growing season. The spread in $B$ results from perturbing soil moisture, but is also directly influenced by the perturbation in radiation and thus $ET_\mathrm{o}$ (Eq. 3 and 5). The ensemble spreads of soil moisture and $B$ are related to each other, and to environmental conditions, more specifically to the relative soil water content (RSW). RSW is defined here as the ratio (root-zone soil water content - wilting point)/(field capacity - wilting point), where the wilting point and field capacity are a function of the spatially varying soil texture.

### 3.3 Showcase 3: Satellite-based Data Assimilation

#### 3.3.1 Background

Most crop DA research aims at parameter estimation, as crop parameters dictate the crop simulation dynamics (Section 2.2). But initial and prior state conditions at each time step also determine crop forecasts, which is the focus of this showcase. The ensemble Kalman filter (EnKF) is the most widely applied method for periodical DA into crop growth models for state updating (Ines et al., 2013) without altering the model structure. The particle filter is also emerging to deal with non-Gaussian error distributions, and to handle multi-model ensembles (Zare et al., 2024), and by extension ensemble trajectories with different (e.g. perturbed) parameters. In any case, the goal is to improve either or both the spatial and temporal variability of



**Table 2.** Ensemble perturbation parameters for showcases (SC) 2 and 3. For the forcings, downward shortwave radiation ($SW$) and precipitation ($P$) are perturbed every hour, with slightly different standard deviations (std) for SC 2 and 3. For the prognostic variables, either soil moisture in the 10 soil compartments ($\theta_k$, $k = 1, \ldots, 10$) or $CC$ and $B$ are perturbed every day, in showcase 2 and 3, respectively. Additive (+) perturbations have a mean of 0 and are drawn from a normal distribution, whereas multiplicative ($\times$) perturbations have a mean of 1 and are drawn from a lognormal distribution. Temporal autocorrelations (tcorr) are applied through a first-order autoregressive model. Since the forcings are perturbed every hour, the tcorr needs to be at least 24 h to obtain meaningful daily aggregated perturbed forcings as input to AquaCrop. Perturbations to prognostic state variables are not cross-correlated with forcing perturbations. SC2 and SC3 use different state perturbations.

| | type | mean | std | tcorr | cross-correlations with other perturbations | | | | | | | | SC2 | SC3 |
| | | | | | $SW$ | $P$ | $\theta_1$ | $\theta_2$ | $\theta_3$ | $\ldots$ | $\theta_{10}$ | $CC$ | $B$ | | |
|---|---|---|---|---|---|---|---|---|---|---|---|---|---|---|---|
| $SW$ | $\times$ | 1 | 0.3; 0.4 | 24 h | 1 | -0.80 | | | | | | | | * | * |
| $P$ | $\times$ | 1 | 0.5; 0.6 | 24 h | -0.80 | 1 | | | | | | | | * | * |
| $\theta_1$ | + | 0 | 0.0060 m³.m⁻³ | 0 days | | | 1 | 0.6 | 0.4 | $\cdots$ | 0.0 | | | * | |
| $\theta_2$ | + | 0 | 0.0055 m³.m⁻³ | 0 days | | | 0.6 | 1 | 0.6 | $\cdots$ | 0.0 | | | * | |
| $\theta_3$ | + | 0 | 0.0050 m³.m⁻³ | 0 days | | | 0.4 | 0.6 | 1 | $\cdots$ | 0.0 | | | * | |
| $\vdots$ | + | 0 | $\vdots$ | 0 days | | | $\vdots$ | $\vdots$ | $\vdots$ | $\ddots$ | $\vdots$ | | | * | |
| $\theta_{10}$ | + | 0 | 0.0015 m³.m⁻³ | 0 days | | | 0.0 | 0.0 | 0.0 | $\cdots$ | 1 | | | * | |
| $CC$ | + | 0 | 0.01 m².m⁻² | 0 days | | | | | | | | 1 | 0.5 | | * |
| $B$ | + | 0 | 0.01 t.ha⁻¹ | 0 days | | | | | | | | 0.5 | 1 | | * |

(unobserved) biomass, irrigation, or yield estimates over what a model-only simulation can achieve. The hope is that state updating will compensate for spatiotemporal errors in meteorological and parameter input that cumulate in the state memory. The success of crop DA varies widely in the literature (Pauwels et al., 2007; de Wit and van Diepen, 2007; Nearing et al., 2012; Lu et al., 2021, 2022) and depends on (i) the coupling strength between observed and unobserved variables, and how accurately the coupling is represented in the model, (ii) the timing of DA, and (iii) the quality of observations and the DA system. For

example: some crop models are unable to adequately propagate surface soil moisture observations to root-zone soil moisture that is essential for crop growth; soil moisture DA is critical in water-limited situations but might have little impact otherwise; and LAI is more directly related to yield during some crop stages (e.g., grain filling) than others.

The AquaCrop state variables are $CC$, $B$, and soil moisture in all compartments ($\theta \in \boldsymbol{\theta}$). In addition, salinity and fertility are part of the system state. These state variables retain information about past interactions and events such as water stress

in the crop system. Depending on the assimilated observation type, a subset of (observable) state variables can be updated through DA, and the other variables will follow the update through model propagation. The cumulative $B$ is best considered as a separate state variable, even if an update to $CC$ propagates to an update in $\Delta B$ via model propagation, that is, the second term in Eq. 5. However, those updates to $B$ through $\Delta B$ are strongly limited by the assimilation frequency. Updating the cumulative $B$ in AquaCrop is essential to improve yield, because $B$ is more directly related to yield than $CC$ (Jin et al., 2020).



Unlike many other models, AquaCrop thus computes its canopy development using $CC$ as a state variable rather than LAI. Whereas $CC$ is strictly limited to [0,1], the upper value of LAI depends on the crop. Therefore, LAI DA can possibly introduce larger updates in LAI (Ines et al., 2013) in other crop models than what is possible with $CC$ for AquaCrop. The consequences thereof for the estimation of biomass and yield need to be further analyzed, because the model pathways from LAI or $CC$ to yield differ. Except for Lu et al. (2021, 2022) who assimilate in situ FCOVER data, no studies have reported the potential of
satellite-based FCOVER retrievals for state updating in AquaCrop.

### 3.3.2    Experiment

In this showcase, we investigate the potential of high-resolution satellite-based FCOVER DA for the estimation of winter wheat biomass and yield, over the Piedmont region in Italy, Europe. Three simulations are performed: deterministic model-only, ensemble model-only (also called open loop, OL), and DA.

AquaCrop is run at 1/112° (~900 m) resolution with crop-specific parameters for winter wheat in GDD mode, using the default conservative parameters offered by the AquaCrop database, and cultivar-specific parameters and fertility stress parameters (management) that are manually calibrated in Lanfranco (2025). The crop parameters are summarized in Appendix C. The calibration is done by minimizing the mean difference between simulated and observed yields from the Global Yield Gap Atlas (GYGA, 2021). This way, the sowing date is set at 20 October, $CC_{\mathrm{x}} = 0.96$ [-], and the soil fertility stress is 40 % on $B$, with
a 5% effect on $CC_{\mathrm{x}}$. High-resolution spatially distributed soil texture information is taken from Geoportale Piemonte (2025d) (1:50,000), and the topsoil texture is used for the entire profile. The latter source has been used to set the domain boundaries. The meteorological forcings are bilinearly interpolated from the forecasts of the fifth generation of atmospheric reanalysis by the European Center for Medium Range Weather Forecasts (ERA5; Hersbach et al., 2020). A deterministic run with this setup for one simulation year over 19,301 crop grid cells takes about 6 min, when run on 12 cpus.

For the ensemble OL, 24 ensemble members are generated by perturbing the forcings similarly as in showcase 2, but unlike showcase 2, the soil moisture state variables are not perturbed, and the state variables $CC$ and $B$ are perturbed instead, as shown in Table 2. In addition, each ensemble member $j$ is assigned a different $CC_{\mathrm{x},j}$ parameter (time-invariant), evenly spaced between 0.92 and 1 around the calibrated center value of 0.96 [-], and the sowing date is varied in response to the perturbed precipitation. The latter ensures enough forecast spread and weakens the model constraints, but none of the crop
parameters is updated. Sowing can happen from 4 October onward when at least 15 mm rain falls in 4 days or less, and this must happen twice. The design of the ensembles has a strong impact on both the skill of the OL and DA, and is further discussed in Lanfranco (2025). An OL run for one simulation year over 19,301 crop grid cells takes about 30 min, when run on 36 cpus.

For the DA, we assimilate 10-day 1/336° (~300 m) CGLS FCOVER observations into the 1/112° (~900 m) gridded ensemble AquaCrop simulations for the years 2017 through 2023. The FCOVER observations are masked using yearly
high-resolution crop maps, i.e. the 10-m High Resolution Layer Croplands product of Copernicus for 2017 through 2020 (CLSM HRL Crops, 2025) and detailed parcel-level regional crop maps of the Piedmont region (1:2000) for 2021 through 2023 (Geoportale Piemonte, 2025a, b, c). If at least 70% of the area within a ~300 m FCOVER pixel is covered with winter wheat according to the crop map, then the FCOVER observation is kept. These masked ~300 m FCOVER observations are





then averaged to ∼900 m for assimilation. The FCOVER observation error standard deviation is set to 0.087 [-], representing

the mean error documented in the CGLS product across the wheat fields and time periods considered for assimilation. When
FCOVER observations are available (every 10 days) at day $i$, each ensemble member $j$ of the AquaCrop crop state is updated
by the EnKF:

$$
\begin{bmatrix} \widehat{CC} \\ B \end{bmatrix}_{j,i}^{+} = \begin{bmatrix} \widehat{CC} \\ B \end{bmatrix}_{j,i}^{-} + \mathbf{K}_i(\text{FCOVER} - \widehat{CC}^{-})_{j,i} \tag{11}
$$

where $\widehat{(.)}^{-}$ refers to the forecast or prior state estimate, and $\widehat{(.)}^{+}$ to the analysis or posterior estimate. The difference $[\text{FCOVER} - \widehat{CC}^{-}]_{j,i}$ is called the 'innovation' and is not a priori bias corrected, in line with earlier LAI DA studies (Albergel et al., 2020; Scherrer et al., 2023). The ensemble innovations are mapped to analysis increments for $CC_{j,i}$ and $B_{j,i}$ through the Kalman gain $\mathbf{K}_i$. The $\mathbf{K}_i$ is derived from ensemble forecast error (co-)variance matrices and the set observation error variance. The increments are limited to what is physically possible for each ensemble member given its own parameter constraints. Specifically, $\widehat{CC}_{j,i}^{+}$ cannot exceed $CC_{\text{pot,sf},j,i}$ (which depends on the perturbed $CC_{\text{x},j}$ parameter). Note that soil moisture is also part of the state, but it is not perturbed or directly updated here. A DA run for one simulation year over 19,301 crop grid cells takes about 45 min, when run on 36 cpus.

The deterministic, OL, and DA runs are evaluated for the times and grid cells with wheat during the years 2017-2023. For the OL and DA simulations, the ensemble mean output is evaluated. The 10-day averaged $CC$ estimates are compared with the assimilated CGLS FCOVER to check internal consistency, and the 10-day averaged $\Delta B$ estimates are evaluated against CGLS DMP retrievals, both only during the growing season. Only positive $\Delta B$ values are retained. Furthermore, yield estimates are evaluated against in situ data of end-of-season yield, using regional survey data from RICA (2025) after aggregation to the municipality level. These yield data are based on individual fields and cover only a small fraction of all wheat fields that are modeled (and for which FCOVER data are extracted).

## 4 Results and Discussion

### 4.1 Showcase 1: Crop Parameterization

Figure 4 shows an example of AquaCrop $CC$ and $\Delta B$ time series, obtained with a generic crop parameterization either in calendar or in GDD mode, along with CGLS reference data for a location in East Europe. The simulations in GDD mode use parameters constrained by satellite information (VIIRS-GLSP). At this location, both crop parameterizations result in good AquaCrop time series of 10-day averaged $\Delta B$ compared to CGLS DMP, but the parameterization in GDD strongly outperforms the parameterization in calendar days in terms of $CC$ compared to CGLS FCOVER data: the timing, shape and interannual variability of $CC$ in GDD are more in line with the reference FCOVER data compared to the calendar day run, indicating that the timing of the crop stages is improved and the impact of stresses is better captured, as seen in the year 2018. Since the $CC_{\text{x}}$ parameter is not calibrated in either approach, there is a trivial absolute bias (Appendix B).





The four left panels of Figure 5 show multi-year averaged $CC$ and $\Delta B$ for both crop parameterizations, for the maximal
growing period defined in Section 3.1. The averaged $CC$ is higher for the crop in calendar days (fixed growing season) than in
GDD, and it has a latitudinal pattern that responds to the temperature pattern. The $CC$ averages are much lower for the crop
in GDD, because the growing season is generally shorter and reduces towards the south (more available heat units). The right
panels of Figure 5 show the time series correlation between the AquaCrop simulations and independent satellite data for all
cropland grid cells in Europe. The simulations with GDD crop parameterization improve $CC$ over almost the entire European
continent, as can be seen in Figure 5d, except for a strong reduction in Norway. In cold regions, the growing season length
computed in GDD (see Appendix B) can vary substantially between years and a median parameter estimate as input may lead
to high errors. In addition, the VIIRS-GLSP product may experience errors due to early or late snow. For $\Delta B$, the spatially
variable GDD crop parameters predominantly improve the simulations in the southern and eastern part of the domain.

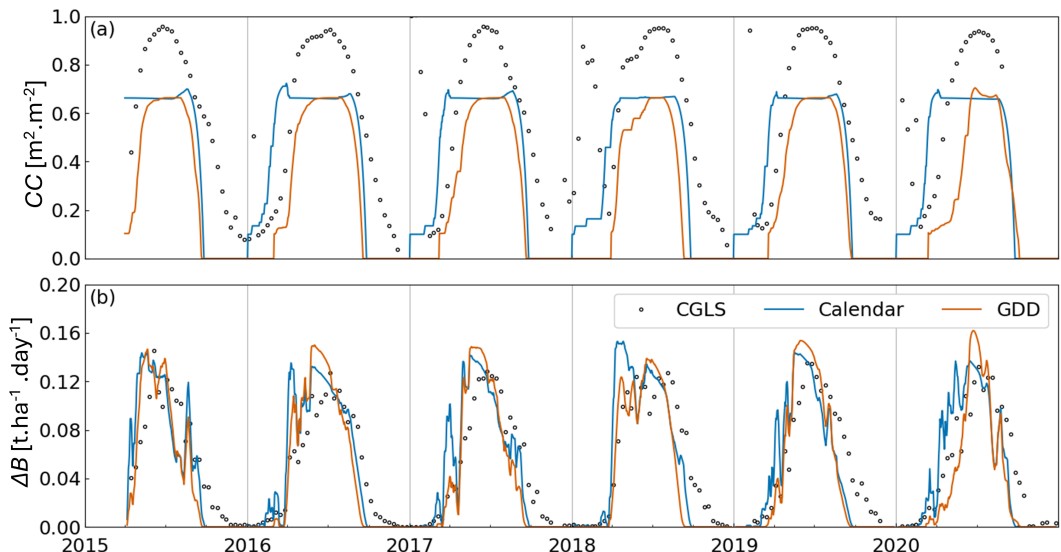

**Figure 4.** Time series of AquaCrop (a) $CC$ and (b) $\Delta B$ for generic crops in calendar and GDD mode, together with reference CGLS
FCOVER and DMP, respectively. The GDD mode uses satellite-based crop parameter constraints. The grid cell location is 46.25 °N, 24.55
°E.



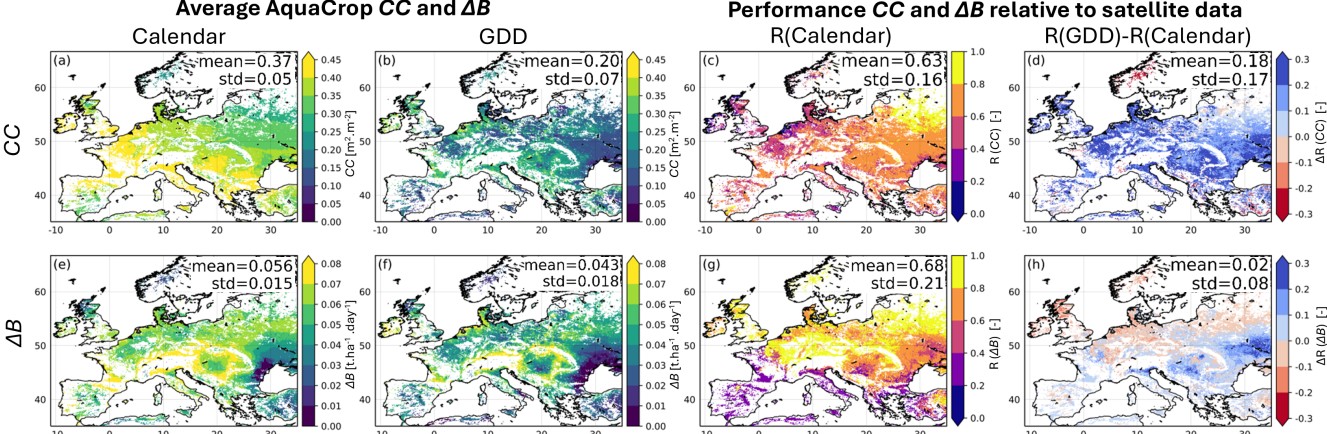

**Figure 5.** (Left) Maps of multi-year averaged AquaCrop (a) $CC$ for a generic crop in calendar days and (b) $CC$ for a generic crop in GDD mode, with satellite-based crop parameter constraints. (e,f) same as (a,b) but for $\Delta B$. (Right) Performance of 10-day AquaCrop (c,d) $CC$ and (g,h) $\Delta B$: time series correlation (R) between simulations and CGLS FCOVER and DMP for a generic crop in (c,g) calendar mode, (d,h) difference in R for simulations with parameters in GDD and calendar days. All panels are computed across the maximal growing period (across both simulations and CGLS data, defined in Section 3.1) of 2015-2020.

## 4.2 Showcase 2: Ensemble Simulations

Figure 6a shows the 3-year average relative soil water content (RSW, defined in Section 3.2) and Figure 6b the ensemble standard deviation (spread, uncertainty) in soil moisture over the 0-100 cm profile. The ensemble spread in soil moisture is often higher in regions with low average RSW that indicate water-limited conditions (e.g. Spain). The associated spread in $B$ is shown in Figure 6c and computed at the end of the growing season when the ensemble mean $B$ is maximal. More specifically, the ensemble standard deviation is computed for the percentage deviations from the end-of-season ensemble mean $B$ for each

405    year, and the average spread across the 3 years is shown. Regions with a high average relative $B$ spread are often, but not exclusively, associated with water-limited conditions (in the absence of simulated irrigation) and high soil moisture spread. However, high soil moisture spread is not the only factor that determines the relative ensemble spread in $B$. The latter is also directly influenced by the ensemble perturbation of radiation (and thus $ET_o$) and the magnitude of maximum biomass, which is highest at intermediate latitudes where the balance between temperature, radiation and water availability is optimal (Figure 5e).

Figures 7a and b further illustrate the sensitivity of $B$ to soil moisture for two locations with different water stress levels, but both on silt loam soils (identical wilting point and field capacity). Here, both the ensemble and temporal sensitivity of $B$ to soil moisture can be seen. The site in Figure 7a experiences minimal water stress, supports high $B$ production, and shows little ensemble sensitivity of $B$ to soil moisture. Furthermore, without water limitations, the interannual variation in $B$ is dominated by the interannual variability in temperature and radiation. In contrast, Figure 7b depicts a southern location where the relative

ensemble $B$ spread is consistently high, as soil moisture frequently approaches the wilting point. Under these conditions, interannual variations in soil moisture also directly translate into interannual $B$ variations. In 2017, insufficient winter soil



moisture recharge severely restricts biomass productivity, and the spread of the ensemble $B$ is large for low $B$ values early in the growing season.

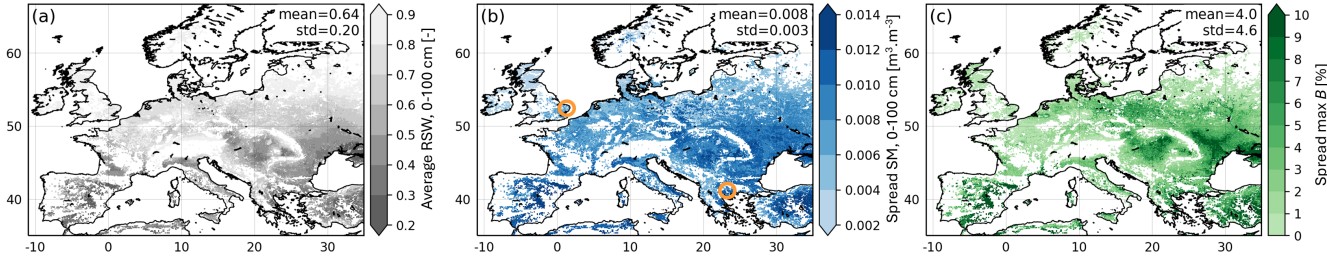

**Figure 6.** Multi-year averages of (a) relative root-zone soil water content (RSW), (b) ensemble spread in root-zone soil moisture (SM), and (c) ensemble spread in end-of-season (max) $B$, for 2015-2018. The centers of the circles in panel (b) mark the locations for Figure 7.

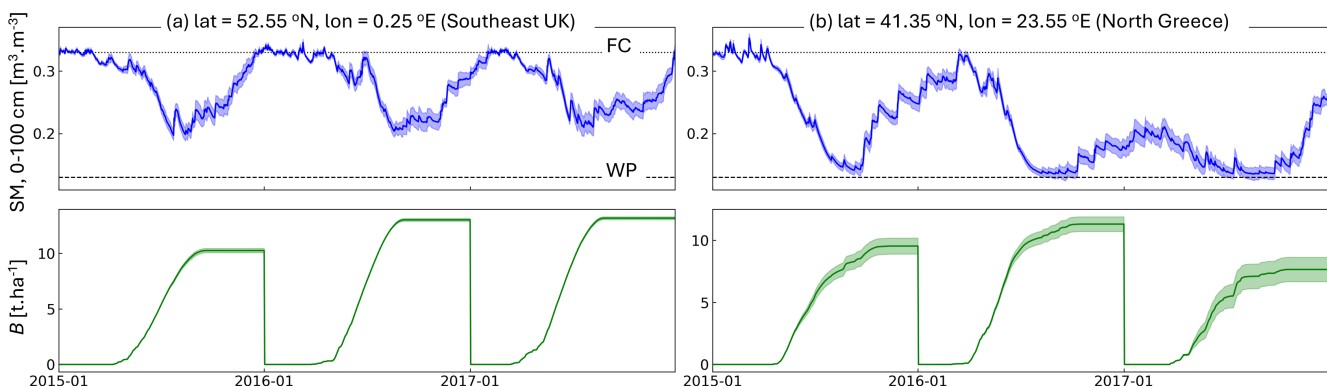

**Figure 7.** Time series of (line) ensemble mean root-zone soil moisture (SM) and biomass, along with (shading) their ensemble spread, for a location (a) without water limitations and (b) with water limitations, both marked on Figure 6b. The lines for FC and WP refer to the location's water contents at field capacity and wilting point.

### 4.3 Showcase 3: Satellite-based Data Assimilation

The average 2017-2023 yield values observed for winter wheat per municipality in the Piedmont area are shown in Figure 8a, along with the number of samples for each value. The samples come from different fields (locations and sizes not disclosed) within the municipality each year, and are not collected every year (Figure 8c). The goal of this showcase is to steer AquaCrop yield simulations to reference yield values by assimilating completely independent FCOVER satellite data, when and where wheat is present. The number of assimilated FCOVER observations per grid cell is shown in Figure 8b. Since the availability

of satellite observations is rather uniform, the number of assimilated observations here reflects how often a ∼900-m grid cell contains wheat fields during the 7-year study period.




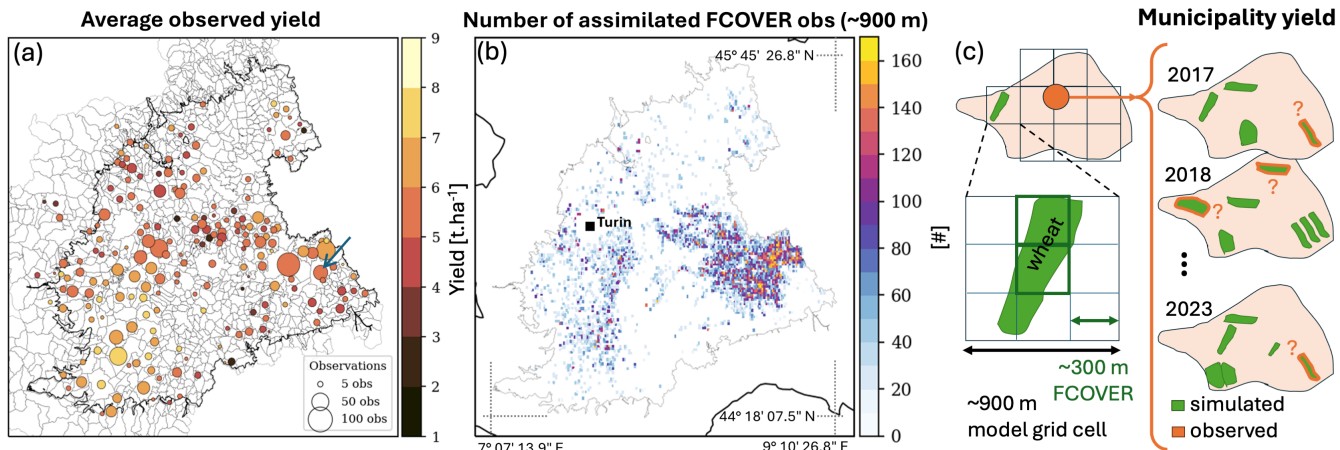

**Figure 8.** (a) Average yield reference values for winter wheat per municipality, for 2017-2023. The dot size represents the number of samples in time and space within one municipality. (b) Total number of assimilated FCOVER retrievals for each ∼900 m model grid cell. The central outline in both plots delineates the Piedmont study area. Panel (a) also shows municipalities, with an indication of the municipality overlying the grid cell in Figure 9, and panel (b) shows the borders of Italy. (c) Sketch of the various spatial supports of gridded modeling, satellite observations, and reference yield data at the municipality level. The green polygons are wheat fields according to crop maps. The number of fields contributing to the each yearly yield estimate is known, but locations are not.

Figure 9a shows time series of $CC$ for the deterministic, ensemble OL, and DA simulation, along with the assimilated FCOVER observations, for a single model grid cell within the municipality of Tortona, Province of Alessandria (marked on Figure 8). Note that the location of the wheat fields within this grid cell differs every year and the FCOVER data are thus extracted over different parts of the model grid cell. For the deterministic run, the sowing date is fixed to 20 October, and the yearly maximal $CC$ is close to $CC_{\mathrm{x,sf}}$, i.e. $CC_{\mathrm{x}}$ (0.96) modulated by a fertility stress effect of 5%, with little interannual variability. This means that there is little variation in water stress across the years (winter-spring) at this location. The growing season length varies with the number of heat units (GDD). The OL $CC$ simulations are often lower than the deterministic ones, because (i) the dynamic sowing criterion delays the start of the growing compared to the deterministic run which assumes a (too early, homogeneous) fixed sowing date on 20 October, and (ii) the perturbations to $CC$ cannot exceed $CC_{\mathrm{pot,sf}}$, leading to a perturbation bias. (In some locations, the OL can advance the sowing date, which leads to a slightly higher $CC$ at the beginning of the growing season.) Both the deterministic and OL simulations often track the independent FCOVER observations very well. At this location, the model overestimates $CC$ in 2020, 2021 and 2022. The DA pulls the simulations towards the FCOVER observations, as intended. However, from senescence onward, the $CC$ estimates no longer depend on values of the previous time step (Appendix A), and DA updates to $CC$ do no longer propagate in time.

The $\Delta B$ estimates are derived from daily differences of fully updated $B$ time series and are shown for the various simulations in Figure 9b. Again, the precipitation-dependent sowing criterion in the OL corrects the early erroneous $\Delta B$ that are simulated by the deterministic run with a fixed sowing date at this location. In the DA, the $\Delta B$ nicely responds to $CC$ updates, e.g. by




pushing the high productivity period to a later time in 2020. In doing so, the DA output corresponds better with the independent

DMP reference data (not assimilated).

Figure 9c shows the simulated yield time series for the single model grid cell, and the observed values for the entire overlying municipality. The deterministic simulation shows very little variation across the years (discussed below) and the OL yield is consistently lower than the deterministic simulation at this location. FCOVER DA only has a small impact on yield estimates, and the variation introduced through DA does not better align with that of the reference yield data. However, the reference

yield data pertain to varying fields over the years, and can thus not be directly compared to the yield simulations nor to the assimilated FCOVER observations for a single model grid cell in Figure 9a.

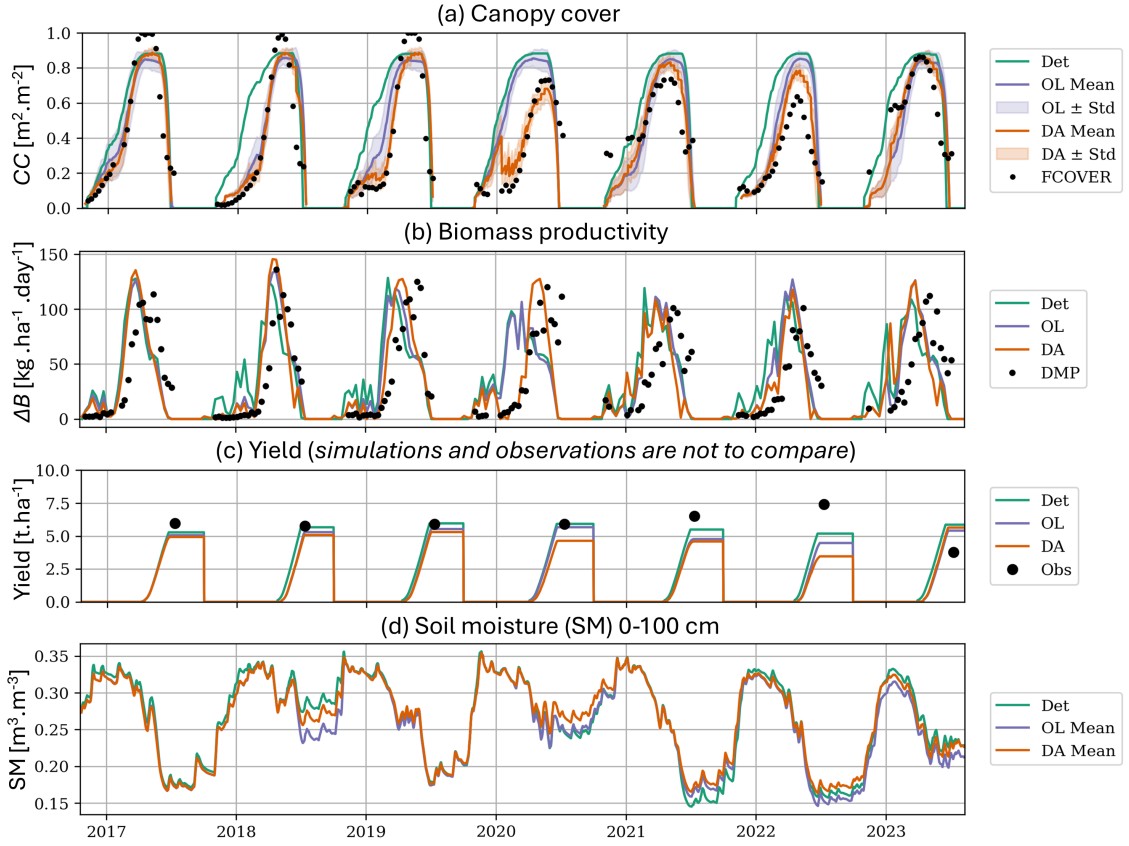

**Figure 9.** Time series of (a) $CC$, (b) $\Delta B$, (c) yield and (d) root-zone soil moisture (SM, smoothed with a 7-day window), for a grid cell within the municipality of Tortona, Province of Alessandria (centered at 44°54'16.6"N, 8°47'39.8"E). Note that the wheat fields within this grid cell rotate every year. The lines refer to deterministic (Det), ensemble mean OL and DA simulations, with the shading reflecting the OL and DA ensemble spread. The dots refer to (a) assimilated 10-day FCOVER retrievals extracted over the wheat field(s) of each year, (b) independent 10-day DMP retrievals, and (c) in situ yield data for the entire municipality (marked by a blue arrow in Figure 8a, covering many other model pixels in addition to the single one selected to plot model results).





A summary of the spatiotemporal performance metrics for the three simulations is given in Table 3. For $CC$, the ensemble OL outperforms the deterministic run in terms of correlation (R), RMSD and bias. This is because the precipitation-dependent varying sowing date is likely more in line with field practices. By design, FCOVER DA further improves $CC$ in all metrics.

Following $CC$, the ensemble OL and DA improve $\Delta B$ over the deterministic run, with the R increasing from 0.54 (and 0.66) [-] for the deterministic (and OL) output to 0.76 [-] for the DA output. For the yield, the deterministic AquaCrop yield estimates only deviate from the in situ observations with a small bias of -0.1 ton.ha$^{-1}$, which is a natural result from the calibration against GYGA data. However, the simulated variation in time and space is very low, leading to a very low R of 0.12 [-] for the deterministic run. This is even further deteriorated in OL simulations (R=0.09) and the DA can only recover some variation

(R=0.13), but at the expense of an increased bias (-0.8 ton.ha$^{-1}$).

**Table 3.** Spatiotemporal performance of the deterministic (Det), OL and DA simulations in terms of R, RMSD and bias, for $CC$ and $\Delta B$ against satellite data, and end-of-season yield against in situ data, across all grid cells (or municipalities for yield) and times with winter wheat during 2017-2023.

|  | $CC$ [m$^2$.m$^{-2}$] | | | $\Delta B$ [kg.ha$^{-1}$.day$^{-1}$] | | | Yield [t.ha$^{-1}$] | | |
| --- | --- | --- | --- | --- | --- | --- | --- | --- | --- |
|  | Det | OL | DA | Det | OL | DA | Det | OL | DA |
| R [-] | 0.66 | 0.75 | 0.86 | 0.54 | 0.66 | 0.76 | 0.12 | 0.09 | 0.13 |
| RMSD | 0.27 | 0.20 | 0.15 | 61.2 | 61.2 | 59.1 | 1.39 | 1.49 | 1.67 |
| Bias | 0.13 | 0.04 | 0.00 | 49.5 | 46.8 | 43.7 | -0.10 | -0.46 | -0.80 |

Figure 10a shows that the simulated variation in OL yield (spatially aggregated over the municipality) across all years and municipalities is low and does not agree with the reference data (R=0.09). DA introduces variability in Figure 10b, leading to a slight improvement in spatiotemporal yield pattern (R=0.13). However, the RMSD and bias increase, because the DA increments are biased negative in much (not all, see Lanfranco (2025)) of the study domain, due to model constraints: $CC_i$

cannot be updated above $CC_{\mathrm{pot,sf},i}$, which is set by uniform crop and fertility parameters, and the yield range is limited by $CC_{\mathrm{pot,sf},i}$ and $HI_{\mathrm{o}}$.

The very low spatiotemporal variation in yield estimates is thus due to parameter constraints, and also a too low sensitivity of yield to environmental conditions (weather, soil). There might also be a model coupling bias (Crow et al., 2024) due to crop (or other) parameter choices that prevent an adequate propagation of $CC$ and $B$ updates to yield updates. Furthermore,

the FCOVER observations might not be sufficiently informative about wheat yield, due to some mixing of other crops in the 300 m data, retrieval assumptions, or because FCOVER retrievals cannot capture some stresses (e.g. stomatal closure due to water stress). Finally, the yield reference data are not representative of the entire municipality: the dot sizes of the scatter plots in Figure 10 highlight that many yield data points are based on just one or a few fields per year in the entire municipality (Figure 8c).





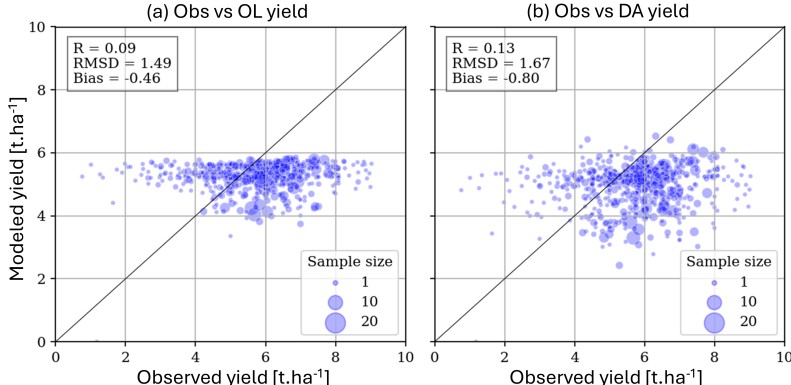

**Figure 10.** Spatiotemporal performance of the OL and DA simulations for end-of-season yield against in situ data, across all municipalities and years with winter wheat in 2017-2023. The dot size refers to the number of individual fields that contribute to the municipality average, for each observed yearly yield estimate.

## 4.4 Pathways Forward

The AquaCrop model alone already offers qualitative simulations of seasonal crop development for generic and specific crops, at both coarse and fine resolutions, when adequate crop parameters are given. However, when parameters related to the sowing date, crop stages, fertility, and maximum $CC_\mathrm{x}$ are fixed, then the interannual and spatial variation in crop simulations are limited. Satellite observations can be used to a priori determine spatially varying crop parameters (showcase 1) or to sequentially update the in-season crop state (showcase 3) and parameters, and thereby improve spatiotemporal patterns in crop simulations. However, future research is needed to optimize crop modeling and DA.

First, showcase 3 shows that crop state updates through satellite DA are beneficial but limited by parameter constraints. For example, crop growth can be advanced or delayed through DA, but the dates to switch to another crop stage remain unaltered, and the updated $CC$ cannot exceed $CC_\mathrm{pot,sf}$. Future work should focus on adapting the timing of the stages to updated $CC$ levels, i.e. to make the crop stages truly state-dependent, rather than precomputing them based on knowledge of the temperature record. Another option is to update (a subset of) the crop and fertility parameters along with the state during the DA: more variation in parameters could result in improved spatiotemporal variation of crop simulations, and could support more state update flexibility. Furthermore, parameter updating could possibly improve the model coupling, i.e. the propagation of updates from observable model variables (e.g. $CC, \theta_1, \ldots$) to (unobserved) variables, such as yield. To keep each model trajectory self-consistent in the presence of the strong model structural constraints set by AquaCrop, particle filtering might be recommended over EnKF for joint parameter and state updating. This option is available in LISF, but not yet tested with AquaCrop.

Second, a crucial aspect of satellite DA is the characterization of forecast and observation errors. In showcases 2 and 3, we introduced perturbations to create ensemble forecasts, guided by iterative study and expert insights, but other options should be explored. By optimizing forecast perturbations and observation errors (e.g. varying in space and time), the effectiveness of DA can be enhanced. Lanfranco (2025) shows how removing perturbation bias correction and assigning distinct parameters to





individual ensemble members, as in showcase 3, causes ensemble OL simulations to deviate more from the deterministic ones. Furthermore, given the bounded nature of key variables such as $CC$, an analysis of ensemble medians rather than means as simulation output can be considered.

Third, showcase 3 only updates crop state variables directly, and soil moisture follows through model propagation (Figure 9). Soil moisture is excluded from the update vector due to the lag between soil and crop responses. However, soil moisture updating can correct for water stresses and biomass will respond to this (as suggested by showcase 2). A joint update of soil moisture and crop variables through multivariate and multi-sensor DA (Heyvaert et al., 2024) might further improve estimates of the entire crop-water system. Additionally, satellite DA can aid in estimating other variables, such as irrigation (Massari et al., 2021; Busschaert et al., 2024; Corbari et al., 2025).

Fourth, our study uses satellite data, whereas most previous studies have relied on field observations to improve crop simulations. The satellite retrievals may not accurately capture the spatiotemporal variation in crop conditions, due to e.g. assumptions in the retrieval algorithm, crop classification errors, or insensitivity to some stresses (e.g. stomatal closure due to water). Future work can explore other or higher resolution data, e.g. from Sentinel-2.

Finally, the evaluation data are prone to errors. As noted, satellite retrievals are not perfect, but neither are yield or other in-field reference data. Figure 8 and 10 highlight that some yield data are based on very few samples from varying fields of different sizes, and these might not be representative for grid cell estimates that are aggregated to the municipality level. A systematic collection and distribution of field data would help the development and tuning of future crop DA systems.

## 5 Conclusions

Crop modeling and data assimilation can be boosted by exploiting the increasing availability of compute power and satellite data. By incorporating AquaCrop v7.2 into the NASA LISF v7.5, it is possible to run AquaCrop at any spatial resolution, with a range of different input sources for meteorology, soil and crop parameters. Furthermore, the system facilitates producing ensembles to obtain uncertainty estimates of crop variables, and it gives access to tools for satellite data assimilation. This opens unprecedented opportunities for regional biomass, yield and irrigation estimation. However, crop models are often structurally less flexible than the land surface models that are typically used within the LISF, and therefore a community-based effort is needed to advance crop data assimilation.

In this paper, we present three exploratory applications of AquaCrop v7.2 within LISF v7.5, employing different study domains, spatial resolutions, forcings, perturbation settings and other input. First, the potential of using satellite information to constrain spatially distributed generic crop parameters is illustrated for coarse-scale simulations of canopy cover and biomass over Europe. Compared to a fixed calendar-day parameterization, spatially variable GDD growth stage parameters consistently improve simulations of canopy cover for a generic crop over all of Europe, and biomass simulations improve for much of the southern and eastern parts of Europe.

Second, ensemble simulations are created by perturbing meteorological input and soil moisture state variables for coarse-scale simulations over Europe, using the satellite-constrained generic crop parameterization in GDD. These ensembles confirm





that biomass uncertainty is most sensitive to uncertainties in root-zone soil moisture in water-limited regions. This is in line
with the expectation (by design) that for these regions, the interannual variation in soil moisture determines the temporal
variation in biomass estimates. These findings allow to anticipate what soil moisture data assimilation may (or may not) offer
to improve biomass in upcoming studies.

Third, high-resolution satellite-based FCOVER data assimilation is performed over the Piedmont region in Italy, focusing on
winter wheat fields. The goal is to improve spatiotemporal estimates of canopy cover, and the unobserved biomass and yield.
Through FCOVER assimilation, the canopy cover is improved by design, and the intermediary biomass is also improved.
FCOVER assimilation also slightly improves the yield estimates, but the impact is very small. The model on its own already
performs well in terms of mean absolute yield values, but performs poorly in spatiotemporal variability. Data assimilation can-
not much improve the latter because of strong model constraints related to the timing of the crop stages, the maximum canopy
cover, fertility and harvest index as parameters. Furthermore, the yield reference data pertain to (at most a few) individual
fields which vary in time and space and are hard to directly compare to gridded model or data assimilation estimates. More
assimilation impact can be expected by making the model crop stages state-dependent, by joint state and parameter updating,
or by assimilating higher resolution satellite data.

Despite their continental coverage (showcases 1 and 2), or high resolution (showcase 3), all showcase experiments are
completed in a few hours of walltime, when parallelized on a high-performance Linux cluster. The computational efficiency
of NASA LISF and the open-source AquaCrop model are encouraging to advance regional-scale and high-resolution crop
modeling and DA. This efficiency enables future multi-sensor and multi-variate assimilation to effectively constrain the soil
water, fertility, and vegetation components of crop models.

*Code and data availability.* The AquaCrop code is available at https://github.com/KUL-RSDA/AquaCrop under a BSD-3 license. The ver-
sion 7.2 of the model used for this paper is archived on zenodo with doi 0.5281/zenodo.17140665 (De Lannoy et al., 2025). The imple-
mentation within NASA's LIS is available at https://github.com/NASA-LIS/LISF under an Apache-2.0 license. The LIS input for the three
showcase experiments is provided on zenodo with doi 10.5281/zenodo.17141268 (Bechtold et al., 2025).





## Appendix A: AquaCrop Canopy Development

The canopy development in AquaCrop is determined by piecewise functions, described below. The same functions are used to calculate the potential $CC_{\mathrm{pot},i}$, the potential $CC_{\mathrm{pot,sf},i}$ with fertility stress only (no water stress), and the actual $CC_i$ [m$^2$/m$^2$]
which accounts for all stresses. However, $CC_{\mathrm{pot},i}$ is computed with time-invariant parameters $\boldsymbol{\alpha}_{\mathrm{cc}} = [CC_{\mathrm{o}}, CC_{\mathrm{x}}, CGC, CDC,$
times to crop stages], whereas potential $CC_{\mathrm{pot,sf}}$ uses time-variant parameters $\boldsymbol{\alpha}_{\mathrm{cc},i}(CC_{i-1}, \boldsymbol{\theta}_i)$ that are adjusted for fertility stress only, and the actual $CC_i$, uses time-variant parameters $\boldsymbol{\alpha}_{\mathrm{cc},i}(CC_{i-1}, \boldsymbol{\theta}_i)$ that are adjusted for water, fertility and salinity stresses and $CC_{i-1}$ until senescence. If fertility stress is set, then $CC_{\mathrm{pot},i}$ without any water stress is effectively constrained to $CC_{\mathrm{pot,sf},i}$, which is determined by (among others) the time-variant $CC_{\mathrm{x,sf},i}$ parameter. Specifically, fertility stress is used (i)
to rescale (increase or decrease) the maximum attainable $CC_{\mathrm{x}}$ parameter to $CC_{\mathrm{x,sf},i}$, (ii) to determine the decrease in $CC_{\mathrm{x,sf},i}$ during the mid-season stage, and (iii) to adjust $WP_i$ (Eq. 5).

The stage-dependent piecewise function for actual $CC_i$ (determined by e.g. $CC_{\mathrm{x},i} \in \boldsymbol{\alpha}_{\mathrm{cc},i}$) is defined as follows:

$$CC_i = f_{\mathrm{cc}}(i, \boldsymbol{\alpha}_{\mathrm{cc},i}(CC_{i-1}, \boldsymbol{\theta}_i)) = \begin{cases} CC_{\mathrm{o},i}\, e^{CGC_i\, \Delta t_{0 \to i}} & CC_i \leq CC_{\mathrm{x},i}/2, \text{ and } t_0 \leq i \\ CC_{\mathrm{x},i} - 0.25 \frac{CC_{\mathrm{x},i}^2}{CC_{\mathrm{o},i}} e^{CGC_i\, \Delta t_{0 \to i}} & CC_{\mathrm{x},i}/2 < CC_i \leq CC_{\mathrm{x},i} \\ CC_{\mathrm{x},i} & t_{\mathrm{ccx}} \leq i < t_{\mathrm{se}} \\ CC_{\mathrm{x},i}\, (1 - 0.05(e^{\frac{3.33 CDC_i}{CC_{\mathrm{x},i}+2.29} \Delta t_{\mathrm{se} \to i}} - 1)) & t_{\mathrm{se}} \leq i \end{cases} \tag{A1}$$

with $t_0$ indicating the day of crop emergence (defined as input parameter $\in \boldsymbol{\alpha}_{\mathrm{cc}}$), $t_{\mathrm{ccx}}$ representing the first day on which
$CC_{\mathrm{x},i}$ is achieved (diagnosed during the simulation), and $t_{\mathrm{se}}$ the day when senescence sets in (defined as input parameter). The expression $\Delta t_{. \to i}$ refers to the number of days from a specific crop stage to the current time $i$.

The first two function pieces describe the crop growth from emergence up to maximum canopy, here marked as $CC_{\mathrm{x},i}$. However, depending on whether we compute $CC_{\mathrm{pot},i}$, $CC_{\mathrm{pot,sf},i}$, or actual $CC_i$, the maximum canopy refers to either the given parameter $CC_{\mathrm{x}}$, its equivalent $\mathrm{CC}_{\mathrm{x,sf},i}$ modulated by fertility stress alone, or its equivalent modulated by water, fertility
and salinity stresses $CC_{\mathrm{x},i}$. The same holds for the $CGC_i$ (and later $CDC_i$) parameter. The third piece covers the period between reaching $CC_{\mathrm{x},i}$ until the onset of canopy senescence. The fourth piece represents the canopy decline after senescence until maturity and the end of the season.

For the first three pieces, the parameters $\boldsymbol{\alpha}_{\mathrm{cc},i}$ are scaled by $CC_{i-1}$, so that $CC_i$ is effectively based on memory of the previous vegetation state. However, for the last piece, the parameters are only rescaled by $CC_{\mathrm{se}}$, i.e. $CC_i$ at the time of
senescence, and $CC_i$ forecasts no longer depend on $CC_{i-1}$ values, soil moisture or fertility conditions. Again, for $CC_{\mathrm{pot},i}$, the canopy development is described by the same functions as Eq. A1, but the parameters $\boldsymbol{\alpha}_{\mathrm{cc}}$ are independent of water or fertility conditions, i.e. $CC_{\mathrm{pot},i} = f_{\mathrm{cc}}(i, \boldsymbol{\alpha}_{\mathrm{cc}})$ (Eq. 2).



## Appendix B: Crop Parameterization Using GLSP Satellite Data

Showcase 3 uses satellite data to estimate crop parameters related to crop stages, growth and decline. Only one crop season
per calendar year is considered. The first full crop season within the calendar year is extracted from the GLSP product (Zhang
et al., 2022). The GLSP dataset provides estimates of crop stages (A-F in Figure 3) in day of the year (DOY) that need to be
converted to growing degree day (GDD), if AquaCrop is run in GDD mode.

Due to a lack of information on sowing or planting, the simulations are initialized on the first of January each year, and
the GDD value for each crop stage is computed relative to this start date for simplicity. The time in GDD at $CC_o$ is used to
define the time from transplanting to recovered transplant, where the $CC_o$ immediately starts at a value of 0.1 $[m^2.m^{-2}]$ as
a recovered transplant. The GDDs [°C day] for each crop stage are calculated following the default AquaCrop method (Raes
et al., 2025b) by cumulating only positive residual average temperatures after subtracting a crop-specific base temperature
$T_{base}$ [°C], below which crop development does not occur, i.e.

$$GDD = T_{avg} - T_{base} \tag{B1}$$

The average day temperature $T_{avg}$ is the average of that day's maximum and minimum temperature, after limiting the maximum
temperature to the range set by $T_{base}$ and $T_{upper}$. The latter is the upper temperature at which crop development no longer
occurs and it is set to an estimated value of 30 °C. $T_{base}$ is set to a universal value of 5 °C, in line with e.g. Zaks et al. (2007).

Specifically, the DOY values are extracted from the GLSP dataset for each 0.05° grid cell and for the different phenology
stages (A, B, C, D, E) shown in Figure 3. The daily maximum and minimum 2-m air temperature from MERRA-2 is then used
to calculate the cumulative GDD for each AquaCrop crop stage (I to V, see Table 1), starting from the day of transplanting
(which in this study equals emergence). This is done for each individual year, with the MERRA-2 meteorology bilinearly
interpolated to the 0.05° resolution. The median GDD value across the 10 years is then taken to determine the crop stages I
through V as parameters in the crop file.

Next, the $CGC$ and $CDC$ parameters are estimated by inverting the exponential growth function of the first piece of
$f_{cc}(.)$ (Eq. A1), i.e. from emergence to $CC_x/2$, and the exponential decay function of $f_{cc}(.)$ for the period between onset of
senescence (point D in Figure 3) and 50% senescence (point E in Figure 3). It is assumed that the potential $CC_o = 0.1$ and
maximal attainable $CC_x = 0.85$ are given, and that the observed actual timing of the median GLSP stages coincides with the
timing of the unobserved potential timing of the required AquaCrop stages (Figure 3). The potential $CC_o$ and $CC_x$ could be
further calibrated in the future. $CGC$ $[^oC.day^{-1}]$ is found using the AquaCrop function for $CC_i \leq CC_x/2$, with $\Delta t_{0 \rightarrow i}$ the
time between $CC_o$ (at time A in Figure 3) and $CC_i = CC_x/2$ (at time B in Figure 3) in GDDs:

$$CC_i = CC_o e^{CGC \, \Delta t_{0 \rightarrow i}} \rightarrow CGC = \frac{\ln \frac{CC_x/2}{CC_o}}{\Delta t_{A \rightarrow B}} \tag{B2}$$

where the time (in GDDs) to reach $CC_x/2$ is taken from the GLSP dataset (GDDs between point A and B, $\Delta t_{A \rightarrow B}$).





Similarly, for $CDC$ [$^o$C.day$^{-1}$], the canopy decline function is used, with $\Delta t_{se\to i}$ the time between start of senescence ($CC_x$ at time D in Figure 3) and $CC_i = CC_x/2$ (at time E in Figure 3) in GDDs:

$$CC_i = CC_x[1 - 0.05(e^{\frac{3.33 CDC}{CC_x+2.29}\Delta t_{se\to i}} - 1)] \to CDC = \frac{2.40(CC_x + 2.29)}{3.33\,\Delta t_{D\to E}} \tag{B3}$$

where the time in GDDs from the start of senescence ($CC_x$) to reach 50% of canopy decline ($CC_x/2$) is used, indicated by the time in GDDs between point D and E ($\Delta t_{D\to E}$).

Note again that AquaCrop requires estimates of the crop stages for the potential, i.e. unstressed, $CC_{pot,i}$ curve. Here, it is assumed that these potential stages have the same timing as the actual stages observed by the satellite data. The $CC_{pot,i}$ curve is defined by the slopes of $CGC$ and $CDC$, derived above, and the lower and upper thresholds of $CC_o$ and $CC_x$, which are guessed. As the $CGC$ and $CDC$ are derived from satellite-based and thermal data, stresses that could affect growth speed or senescence are inevitably included in the parameter estimates. By taking the median value for the GDD stages, from which $CDC$ and $CGC$ are derived, it is assumed that the most representative stages are used for each grid cell. However, the values for $CC_o$ and $CC_x$ are spatially constant in this study. Initial guesses could potentially be derived from the VIIRS-GLPS dataset: $CC_o$ could be estimated by taking the maximum $CC_i$ over 10 years at the DOY of crop stage I, and $CC_x$ can be approximated by taking the maximum $CC_i$ over the entire growing season. However, these estimates would be strongly affected by stresses, incl. fertility stress, and future research should attempt to optimize effective potential (unstressed) parameter estimates.

## Appendix C: AquaCrop Crop Parameters

Table C1: AquaCrop crop parameters for (i) the generic crop in calendar days (showcase 1), (ii) generic crop with satellite-based constraints in GDDs (showcase 1,2), and (iii) winter wheat in the Piedmont area (showcase 3). * refers to the VIIRS-GLSP crop stages in GDD, as in Table 1.

| | Generic (calendar) | Generic (GDD) | Winter wheat (GDD) |
|---|---|---|---|
| Leafy vegetable crop (1), fruit/grain producing crop (2) | 1 | 1 | 2 |
| Crop is transplanted (0), crop is sown | 0 | 0 | 1 |
| Determination of crop cycle : by calendar or growing degree-days | 1 | 0 | 0 |
| Soil water depletion factors (p) are adjusted by ETo | 1 | 1 | 1 |
| Base temperature (°C) below which crop development does not progress | 5 | 5 | 0 |
| Upper temperature (°C) above which crop development no longer increases with an increase in temperature | 30 | 30 | 26 |
| Total length of crop cycle in growing degree-days | -9 | 3123 | 2694 |
| Soil water depletion factor for canopy expansion (p-exp) - Upper threshold | 0.25 | 0.25 | 0.20 |
| Soil water depletion factor for canopy expansion (p-exp) - Lower threshold | 0.55 | 0.55 | 0.65 |





| | | | |
|---|---|---|---|
| Shape factor for water stress coefficient for canopy expansion (0.0 = straight line) | 3 | 3 | 5 |
| Soil water depletion fraction for stomatal control (p - sto) - Upper threshold | 0.50 | 0.50 | 0.65 |
| Shape factor for water stress coefficient for stomatal control (0.0 = straight line) | 3.0 | 3.0 | 2.5 |
| Soil water depletion factor for canopy senescence (p - sen) - Upper threshold | 0.98 | 0.98 | 0.70 |
| Shape factor for water stress coefficient for canopy senescence (0.0 = straight line) | 3.0 | 3.0 | 2.5 |
| Sum(ETo) during stress period to be exceeded before senescence is triggered | 1000 | 1000 | 50 |
| Soil water depletion factor for pollination (p - pol) - Upper threshold | 0.90 | 0.90 | 0.85 |
| Vol% for anaerobiotic point (SAT - [vol%] at which deficient aeration occurs) | 5 | 5 | 5 |
| Considered soil fertility stress for calibration of stress response (%) | 50 | 50 | 40 |
| Shape factor for the response of canopy expansion to soil fertility stress | 2.35 | 2.35 | 5.48 |
| Shape factor for the response of maximum canopy cover to soil fertility stress | 0.79 | 0.79 | 4.70 |
| Shape factor for the response of crop Water Productivity to soil fertility stress | -0.16 | -0.16 | -2.06 |
| Shape factor for the response of decline of canopy cover to soil fertility stress | 7.82 | 7.82 | 7.82 |
| Minimum air temperature below which pollination starts to fail (cold stress) (°C) | 8 | 8 | 5 |
| Maximum air temperature above which pollination starts to fail (heat stress) (°C) | 40 | 40 | 35 |
| Minimum growing degrees required for full crop transpiration (°C - day) | 10 | 10 | 14 |
| Electrical Conductivity of soil saturation extract at which crop starts to be affected by soil salinity (dS/m) | 2 | 2 | 6 |
| Electrical Conductivity of soil saturation extract at which crop can no longer grow (dS/m) | 12 | 12 | 20 |
| Calibrated distortion (%) of CC due to salinity stress (0 (none) to +100 (very strong)) | 25 | 25 | 25 |
| Calibrated response (%) of stomata stress to ECsw (0 (none) to +200 (extreme)) | 100 | 100 | 100 |
| Crop coefficient when canopy is complete but prior to senescence (KcTrx) | 1.10 | 1.10 | 1.10 |
| Decline of crop coefficient (%/day) as a result of ageing nitrogen deficiency etc. | 0.05 | 0.05 | 0.15 |
| Minimum effective rooting depth (m) | 0.30 | 0.30 | 0.30 |
| Maximum effective rooting depth (m) | 1.00 | 1.00 | 1.00 |
| Shape factor describing root zone expansion | 15 | 15 | 15 |
| Maximum root water extraction (m3water/m3soil.day) in top quarter of root zone | 0.048 | 0.048 | 0.048 |
| Maximum root water extraction (m3water/m3soil.day) in bottom quarter of root zone | 0.012 | 0.012 | 0.012 |
| Effect of canopy cover in reducing soil evaporation in late season stage | 60 | 60 | 50 |
| Soil surface covered by an individual seedling at 90 % emergence (cm2) | 15 | 15 | 1.5 |
| Canopy size of individual plant (re-growth) at 1st day (cm2) | 15 | 15 | 1.5 |
| Number of plants per hectare | 666667 | 666667 | 4000000 |
| Canopy growth coefficient (CGC): increase in canopy cover (fraction soil cover per day) | 0.10368 | 0.10368 | 0.03418 |
| Maximum canopy cover (CCx) in fraction soil cover | 0.85 | 0.85 | 0.96 |
| Canopy decline coefficient (CDC): decrease in canopy cover (in fraction per day) | 0.080 | 0.080 | 0.097 |



| | | | |
|---|---|---|---|
| Calendar Days: from transplanting to recovered transplant | 0 | 0 | 15 |
| Calendar Days: from transplanting to maximum rooting depth | 80 | 80 | 130 |
| Calendar Days: from transplanting to start senescence | 232 | 232 | 220 |
| Calendar Days: from transplanting to maturity | 365 | 365 | 250 |
| Calendar Days: from transplanting to flowering | 0 | 0 | 170 |
| Length of the flowering stage (days) | 0 | 0 | 15 |
| Crop determinancy unlinked with flowering | 0 | 0 | 1 |
| Building up of Harvest Index (% of growing cycle) | 10 | 10 | 100 |
| Building up of Harvest Index starting at sowing/transplanting (days) | 36 | 36 | 76 |
| Water Productivity normalized for ETo and CO2 (WP*) (gram/m2) | 17 | 17 | 15 |
| Water Productivity normalized for ETo and CO2 during yield formation (as % WP*) | 100 | 100 | 100 |
| Crop performance under elevated atmospheric CO2 concentration (%) | 50 | 50 | 50 |
| Reference Harvest Index (HIo) (%) | 85 | 85 | 48 |
| Possible increase (%) of HI due to water stress before flowering | -9 | -9 | 5 |
| No impact on HI of restricted vegetative growth during yield formation | -9 | -9 | 10 |
| No effect on HI of stomatal closure during yield formation | -9 | -9 | 7 |
| Allowable maximum increase (%) of specified HI | -9 | -9 | 15 |
| GDDays: from transplanting to recovered transplant | -9 | A* | 187 |
| GDDays: from transplanting to maximum rooting depth | -9 | 0.7 D* | 918 |
| GDDays: from transplanting to start senescence | -9 | D* | 2080 |
| GDDays: from transplanting to maturity | -9 | F* | 2694 |
| GDDays: from transplanting to flowering | -9 | 0 | 1281 |
| Length of the flowering stage (growing degree days) | -9 | 0 | 198 |
| CGC for GGDays: increase in canopy cover (in fraction soil cover per growing-degree day) | -9 | Eq. B2 | 0.004843 |
| CDC for GGDays: decrease in canopy cover (in fraction per growing-degree day) | -9 | Eq. B3 | 0.004739 |
| GDDays: building-up of Harvest Index during yield formation | -9 | 140 | 1311 |
| Dry matter content (%) of fresh yield | 10 | 10 | 88 |
| Minimum effective rooting depth (m) in first year - required only in case of regrowth | 0 | 0 | 0 |
| Crop is transplanted in 1st year - required only in case of regrowth | 0 | 0 | 0 |
| Transfer of assimilates from above ground parts to root system is NOT considered | 0 | 0 | 0 |
| Number of days at end of season during which assimilates are stored in root system | 0 | 0 | 0 |
| Percentage of assimilates transferred to root system at last day of season | 0 | 0 | 0 |
| Percentage of stored assimilates transferred to above ground parts in next season | 0 | 0 | 0 |



*Author contributions.* G.DL., L.B., S.dR., Z.H., J.M., S.S., M.VdB. and M.B. contributed to the open source Fortran code of AquaCrop v7.0
and higher versions. L.B. and M.B. led the implementation of AquaCrop inside NASA LIS with the support of S.K., D.M., and E.K.. M.B.
performed the research for showcase 1 and 2, with help from S. dR for showcase 1. N.L. performed the research for showcase 3 with support
from L.B.. D.R. is the main AquaCrop developer in Delphi and M.G., P.S., L.H. contributed to the initial AquaCrop code. G.DL. prioritized
the research, provided supervision, managed the funding, and wrote the paper. All authors contributed to the editing of the paper.

*Competing interests.* The authors declare that they have no conflict of interest.

*Acknowledgements.* The implementation of AquaCrop in LIS benefited from help of James Geiger, the original Delphi AquaCrop source
code was co-developed by Elias Fereres. Other technical support was offered by Johan Boon and Samuel Corveleyn. The computer resources
and services were provided by the High Performance Computing system of the Vlaams Supercomputer Center, funded by FWO and the
Flemish Government (incl. Storage4Climate collaborative grant). This work received support from the European Commission, Horizon 2020
Framework Programme (SHui grant no. 773903), project C14/21/057 of KU Leuven, and the STEREO IV project CROPWAVES (SR/00/412)
of the Belgian Science Policy (Belspo). Louise Busschaert is funded by FWO grant 1158423N.



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
