# Peer review of "Advancing Crop Modeling and Data Assimilation Using AquaCrop v7.2 in NASA's Land Information System Framework v7.5"

_EGUsphere, 2025_

## Author Comment (AC3)

We thank Matt McCabe for the timely and constructive comments. We list the comments in bold fonts below and we provide an answer in normal blue fonts, with suggestions for updated text in italic (additions are underlined). The line numbers in our response refer to our updated manuscript (will be uploaded in the next step).

Note that, in addition to addressing the comments below, we reran the simulations for Showcase 1&2 with ERA5 forcings instead of MERRA2 forcings, because the reference evapotranspiration obtained with MERRA2 in LIS is at the high end for crop simulations. The Figures 4 through 7 will thus be replaced for consistency, but without any consequence for the scientific findings.

**A. Comments specific to Page and Line:**

**1. Page 7. Lines 177-178. The statement from "The entire..." to "...simulation year."**

**This statement appears to imply that this condition holds for both GDD and Calendar-day configurations. However, I don't think this is the case when AquaCrop is run in calendar-day mode. It would be helpful if the authors could clarify this distinction and explicitly state that the described behavior pertains only to the GDD configuration.**

Thank you, good catch. We will update the text as follows:

L.178*: "Consequently, when a simulation is run in GDD mode, the entire temperature record of a simulation year is used to precompute the timing of the stages at the beginning of that simulation year. This limitation does not apply when AquaCrop is run in calendar days...*

**2. Page 8. Figure 3.**

**The parameters related to rooting depth evolution (Zn and Zx) did not appear to be defined in the text prior to this figure. It would be helpful to clarify their meaning in the caption (Zn: minimum rooting depth and Zx: maximum rooting depth).**

**In addition, I understand that the crop stage timings derived from the GLSP product correspond to points A through F, and these should align with AquaCrop's crop stages, as they represent equivalent physiological events. However, in Figure 3, the GLSP points did not appear to align temporally with the corresponding AquaCrop's crop stages (with the exception of point A, which matched the (I) time to emergence). Could the authors clarify whether this misalignment is due to the visualization, or whether there is a conceptual reason for the offset?"**

Thanks, indeed we did not explain Zn and Zx in the text and will add this as follows:

L.167*: "... containing the minimal and maximal potential rooting depth parameter, Zn and Zx [cm] respectively, illustrated in Figure 3..."*

Furthermore, we agree that Figure 3 was confusing. We had tried to convey that the actual satellite-based parameters are not necessarily exactly in line with the potential AquaCrop stages, but this was too farfetched and not explained. We will now align the GLSP points with the AquaCrop stage parameters, in line with what was assumed and effectively done in the methodology.

**3. Page 9. Lines 213-215. The statement from "Furthermore, only…." to "…future versions."**

**Are these parameters (i.e., spatially variable vs. uniform parameters) listed anywhere in the manuscript? It would be helpful for readers if these parameters were explicitly stated.**

We will update the text as follows:

L.217: "*Furthermore, except for meteorological input, only the topsoil texture and crops are currently spatially variable input. The specification of groundwater, irrigation and management parameters (provided via original AquaCrop files) and planting/sowing criteria are uniform, but this can easily be addressed in future versions.*"

**4. Page 10. Lines 231-233. The statement from "The most…" to "…and HIo."**

**First, this statement would be stronger if supported by one or more references. Ideally, the authors might cite studies that have conducted sensitivity analyses for AquaCrop and identified these parameters as the most important crop parameters. As it stands, making this claim without supporting evidence or analysis is not fully convincing.**

**Second, there appears to be a typo, GCG should be CGC (Crop Growth Coefficient – *Canopy Growth Coefficient*). The acronym is used as CGC earlier in the manuscript, so this should be corrected here.**

We agree, and this comment in fact revealed that we also had not properly distinguished between two papers by Lu et al. (2021) in our bibfile (LaTeX), with one on sensitivity and one on data assimilation. Thanks also for catching the typo. We will update the text as follows:

L.236: "*The most important crop parameters are related to the length of the different stages of phenological development, CCx, CGC, CDC, and Zx, marked in Figure 3, and HIo. The relative importance of these parameters depends on the crop type and the environment (Vanuytrecht et al., 20214, Lu et al., 2021).*"

**5. Page 10. Lines 250-251. The statement from "The mapping…" to "…in Appendix A."**

**This comment is related to the earlier comment raised for Page 8, Figure 3, but with an additional point that requires clarification.**

**Table 1 clearly explained the relationships between AquaCrop and GLSP crop stages; however, in Figure 3, these parameters did not align on the same points on the x-axis. AquaCrop's crop stages consistently appeared earlier than the corresponding GLSP, except for one stage. It would be helpful if the authors could clarify whether this misalignment is intentional (e.g., due to methodological reasons) or a visualization issue.**

**Additionally, the definition of stage "II. Time to maximum rooting depth" being set at 0.7*D needs further explanation, why was 0.7 chosen specifically? What is the basis for this proportion?**

We agree that this was confusing and will revise Figure 3 to align the stages. See also our response to comment 2. Furthermore, we will update the text as follows, to explain the 0.7 factor.

Table 1: replace "estimated" by "*guess, see Appendix B*".

L.650 (Appendix B): "*Finally, the time to maximum rooting depth is set to 0.7 D based on an expert guess. This is based on the assumption that (i) this stage will not be reached before mid-season*"

*or the time of CCx, (ii) it should occur before senescence, and (iii) roots grow about 1 cm.day$^{-1}$ in warm soils to a general maximum rooting depth of about 1 m (Allen et al., 1998)."*

**6. Page 12. Line 285. The statement "….an ensemble of 24 members."**

**It would be helpful to provide a brief explanation (either here or in an appendix) of how the perturbations to the meteorological forcing and soil moisture state variables generated the ensemble of 24 members. A short description of the perturbation scheme or the combination logic would greatly improve clarity for readers.**

Sure, we will update the text as follows:

L.294: *"Each member corresponds to a model trajectory, resulting from adding small random values with zero mean to some variables, and multiplying other variables by a small random number around 1. One member is left unperturbed (see below). The random number distribution is determined by perturbations parameters (mean, standard deviation, correlation), which are spatially and temporally constant, and summarized in Table 2. The setup is inspired by state-of-the-art land surface data assimilation studies (Kumar et al. 2008, Heyvaert et al 2023). The resulting perturbations are applied hourly to shortwave radiation and precipitation, and daily to soil moisture variables, and all values are kept within physically possible bounds through resampling."*

We will also touch the discussion:

L.524: *"In showcases 2 and 3, we introduced perturbations to create ensemble forecasts, guided by iterative study, expert insights into the AquaCrop model, and land surface data assimilation studies (Kumar et al. 2008, Heyvaert et al. 2023). However, the forecast perturbations and observation errors should be further optimized (e.g. varying in space and time) to enhance the effectiveness of DA."*

**7. Page 12. Lines 290-291. The statement from "A perturbation…" to "…simulation."**

**To clarify the workflow: was the perturbation bias correction applied before generating and running the ensemble members?**

We will better clarify this procedure as follows:

L.303: *"A perturbation bias correction (Ruy et al. 2008) is applied at each time step during the ensemble simulation to keep the soil moisture ensembles centered around the unperturbed deterministic simulation (one member is left unperturbed): even if all perturbations should have a zero mean, nonlinear effects could introduce a bias in the resulting ensemble means. Perturbation bias correction avoids that unintended biases in ensemble soil moisture propagate into the ensemble uncertainty estimates of CC and B."*

**8. Page 12. Lines 294-295. The statement from "…multi-year average" to "…croplands in Europe."**

**Given the description of Section 3.2.2, I was anticipating that the authors might report the multi-year average ensemble standard deviation for CC, alongside biomass and root-zone soil moisture.**

While I recognized that the primary objective of this showcase is biomass (as stated on Page 9, Line 221), the inclusion of root-zone soil moisture, despite it not being explicitly part of the stated objective, suggests that the analysis is not limited strictly to B. Was there a reason that CC was excluded from this uncertainty assessment?

This is a very pertinent question. We will address this topic as follows in the paper:

L.310: *"The spread in CC is not further discussed, because it is primarily limited by the parameterized crop stages, which are identical for all members because the temperature (GDD) is not perturbed."*

We could possibly add a plot and discussion on the CC spread if required, but that does not add much value. Furthermore, we are actively working on this topic (PhD Louise Busschaert): to obtain good ensembles of e.g. irrigation, we need to perturb temperature (and thus crop stage parameters), which results in a better spread of CC. Nevertheless, the perturbation of meteorological input variables in the current study (shortwave radiation and precipitation) already illustrates the importance and impact of accounting for input uncertainty.

**9. Page 12. Lines 297-299. The statement from "The spread…" to "…water content (RSW)."**

**The explanation of the spread in B as directly resulting from perturbations in soil moisture and radiation (through ETo) is conceptually sound - but doesn't it simplify AquaCrop's internal behavior? As a process-based model, AquaCrop includes multiple threshold-driven stress functions and nonlinear feedbacks among canopy cover, transpiration, and biomass. Consequently, the ensemble variability in B likely reflects not only the imposed input perturbations but also the model's inherent stress-response interactions.**

**The statement that "The ensemble spreads of soil moisture and B are related to each other, and to environmental conditions, more specifically to RSW" would benefit from additional mechanistic clarification. Although RSW is a physically meaningful indicator of soil wetness, AquaCrop's stress functions translate this variable into nonlinear physiological responses. Hence, the relationship between B and soil moisture spreads cannot be attributed to RSW alone without considering these embedded physiological thresholds and nonlinear responses. Clarifying this connection would likely make the interpretation more consistent with AquaCrop's process-based structure.**

We fully agree, thanks for the suggestion. This will be addressed as follows:

L.315: "The spread in B results from… *and multiple threshold-driven stress functions and feedbacks among CC, transpiration, and B. For this showcase, we only relate* the ensemble spreads of soil moisture and B to each other, and to environmental conditions, more specifically to the relative soil water content (RSW)."

L.437: "*AquaCrop's stress functions translate soil moisture into nonlinear physiological responses. This implies that the relationship between soil moisture and B spread is nontrivial and varies across both time and space*."

**10. Page 13. Table 2.**

**I could not find where the choice of probability distributions for the ensemble perturbations (normal for additive variables & lognormal for multiplicative variables) was explicitly justified in the manuscript. It would be useful to explain the rationale behind these**

selections, as the assumed distribution directly influences the shape and magnitude of the propagated uncertainty.

It was also a little unclear whether the perturbed parameter values remain within physically plausible ranges. Depending on the perturbation amplitude and distribution, it is possible that the sampled values may fall outside the realistic domain for the study region. Such unrealistic values could introduce artificial spread in the outputs and potentially complicate interpretation of the resulting uncertainty patterns. Clarifying whether constraints, bounds, or validity checks were applied to the perturbed parameters would help readers assess the robustness of the ensemble design.

We indeed took the bridge from land surface modelling to crop modelling a bit too much for granted, and will add more explanation on the ensembles, their choice (based on literature), and their physical bounds. We will also add text on the potential to optimize perturbations. Please see our answer to comment 6.

**11. Page 13. Lines 321-323. The statement from "The cumulative..." to "...assimilation frequency."**

**It is unclear whether this was implemented in your experiment or just mentioned conceptually.**

Indeed, we mentioned this conceptually here, and then also implement it as such in our experiments (Eq. 5). This will also be addressed in comment 14, and be clarified as follows:

L.345: "*A direct update of B will therefore also be included in our experiment below.*"

**12. Page 14. Lines 329-330. The statement from "Except for..." to "...updating in AquaCrop."**

**I believe there are at least two published studies that have reported on the assimilation of satellite-based fractional vegetation cover into AquaCrop for CC state updating. For example:**

**https://www.mdpi.com/2073-4395/11/11/2265**

**https://www.mdpi.com/2073-4395/9/7/404**

**The authors might want to review these works and perhaps revise the statement on Page 14, Lines 329-330.**

Apologies, this was indeed not right, and we will update the text as follows:

L.352: "*Some studies (Dalla Marta et al., 2019, Lu et al. 2021, Ali Saab et al. 2021, Lu et al. 2022) have assimilated in situ or satellite-based FCOVER data at the field scale, but the potential of satellite-based FCOVER retrievals for state updating in AquaCrop across many fields at the regional scale is understudied.*"

**13. Page 14. Lines 346-347. The statement from "moisture state variables..." to "...in Table 2."**

**Was there a reason why soil moisture state variables were excluded from the perturbation setup in Showcase 3? Although Section 4.4 provides justification, specifically the lag between soil and crop responses, it would strengthen the clarity of this section if that rationale were also mentioned here.**

We agree, and will update the text:

L.370: *"Our primary goal is to update vegetation next, and for simplicity soil moisture will be excluded from the control vector, because there is a feedback lag between soil and crop responses (see Section 4.4)."*

**14. Page 15. Equation 11.**

**Equation 11 lists both CC and B as state variables updated by the EnKF. However, as far as I understood, FCOVER was assimilated primarily to update CC. It would be helpful if the authors clarified how B is handled during the update step: is B explicitly updated by the EnKF, or does it adjust only indirectly through model propagation after CC is updated?**

**This point is related to the earlier comment on Page 13, Lines 321-323, and addressing both would improve clarity regarding the DA update mechanism.**

Sure, we will rephrase this as follows, and please also see our answer to comment 11.

L.385: *"When FCOVER observations are available (every 10 days) at day i, the EnKF jointly updates the forecasted AquaCrop CC and B state variables for each ensemble member j:"*

**15. Page 15. Lines 383-387. The statement from "At this..." to "...the year 2018."**

**There is no issue having this qualitative evaluation of AquaCrop estimates - but to make the evaluation more meaningful, you might also include some quantitative error metrics, such as RMSE, bias, or other standard performance measures, to assess the discrepancy between AquaCrop time-series estimates and the independent satellite data. If possible, incorporating these metrics would likely strengthen the evaluation and provide a clearer picture of how well the model performs at coarse scale.**

We agree, and will update the text as follows, and add the correlation, bias and RMSD to Figure 4. The figure will also slightly change, because we will provide the simulations forced with ERA5.

L.412: *"At this location, the time series correlation over the maximal growing period increases from 0.51 to 0.69 for CC, and 0.89 to 0.92 for ΔB. Since the CCx parameter is not calibrated in either approach (Appendix B), there is a trivial absolute bias that is further amplified in GDD due to its more realistic, but lower, early- and late-season CC values. This bias also propagates into the root-mean-square difference (RMSD) metrics."*

**16. Page 16. Line 391. The statement "...responds to the temperature pattern."**

**The authors might consider including a map (either into Figure 5 or placed in an appendix) illustrating the temperature pattern referenced here.**

Sure, we will add the map in Appendix C, and update the text as follows:

L.417: *"The averaged CC is higher for the crop in calendar days (fixed growing season) than in GDD mode, and it has a latitudinal pattern that responds to the temperature pattern, discussed in Appendix C."*

L.667 (Appendix C): *"The effective growing season length for the two generic crops is compared in Figure C1, where the growing season is defined from the first time CC exceeds the initial CCo until CC drops to 0 $m^2.m^{-2}$. Even if the crop in calendar days is intentionally parameterized to*

*potentially span the entire year (de Roos et al. 2021), cold stresses typically prevent growth in the first month(s), effectively limiting the growing season length. The date of senescence is fixed. This results in a latitude-dependent growing season length, both in terms of days and cumulated GDD (Figure C1a, c). This pattern is not seen for the generic crop in GDD mode, which uses satellite-based phenology to parameterize the stages. For the crop in GDD mode, the relative coefficient of variation is higher, i.e. the growing season length is shorter, and the variation across Europe is similar to that for the crop in calendar day mode (Figure C1b, d)."*

**17. Page 16. Lines 397-398. The statement from "For ΔB…" to "….part of the domain."**

**For CC, the manuscript provided possible explanations for the strong performance reduction in Norway, but no similar interpretation was offered for ΔB. The authors might like to discuss some of the potential reasons why simulations using GDD crop parameterization did not yield improvements in other regions of Europe (particularly western and northern areas).**

We agree. The related figure will slightly change, because we will provide the simulations forced with ERA5, and we will further discuss this as follows:

L.424: *"A more realistic CC and onset of B accumulation improves ΔB for much but not all of Europe, due to spatial variability in several factors - related to both the modeling framework and the reference data. A first factor is inaccurate simulation of transpiration, water stress, and other stresses. In this study, we applied vertically uniform soil profiles, and a spatially uniform parameterization of the generic crops (only the crop stages, CGC and CDC vary spatially), introducing spatial modelling errors. Second, the reference DMP dataset is retrieved using many simplifying assumptions, and it represents a mixture of different crop types after aggregation."*

**(B) General Comments:**

**B1. Appendix C. AquaCrop Crop Parameters.**

**What is the difference between the parameter (Total length of crop cycle in growing degree-days) and the parameter (GDDays: from transplanting to maturity)?**

**Based on their physical meaning, they appear to represent the same concept, which suggests they should be assigned the same value. However, in the second column of the table (Generic GDD), the authors gave Total length of crop cycle in growing degree-days a value of 3123 GDD, while GDDays: from transplanting to maturity was assigned F, meaning each had a different value. This differentiation did not appear in the third column (Winter wheat GDD), where both inputs had the same value (2694 GDD). Might be helpful if this were clarified.**

Great catch, many thanks! The crop file is designed in such a way that it is 'backwards compatible', leading to some confusion. The dummy variable that is read in the line "Total length of the crop cycle in growing degree days" is replaced by the variable read in the line "GDDays: from transplanting to maturity". In other words, the first entry is never used. People would only know if they open up the source code.

We will update Appendix C by replacing the dummy variables by "-9", and adding a note in the caption as follows:

*"Note that some entries are listed in the crop file, but not used in AquaCrop v7.2, and therefore set to -9."*

**B2. Showcase 1. Crop Parameterization.**

**The authors described how inputs related to climate, crop, soil, and field management were parameterized (for field management; only one input – soil fertility level). However, there was no explanation of how other inputs, such as those related to irrigation management, groundwater table, or additional field management, were handled? How were these inputs/parameters parameterized, especially given that the simulations were conducted at a coarse scale where substantial spatial variation exists across the study area?**

Indeed, this was missing and will be added as follows:

*L.270: "…. uniform soil fertility stress…. All other boundary conditions are also set uniformly and in a simplistic way, i.e. no irrigation and no shallow groundwater table (no capillary rise)."*

**B3. Showcase 3. Satellite-based Data Assimilation.**

**A simple exploration of the rationale for choosing fractional vegetation cover (FCOVER) as the assimilated variable in AquaCrop i.e. why was FCOVER selected specifically, rather than other variables - might prove useful here.**

Sure, the text will be updated as follows:

L.328: *"Satellite-based crop DA studies typically use retrievals of leaf area index (LAI), surface soil moisture, the ratio of actual evaporation to potential evapotranspiration, or a combination thereof (Vazifedoust et al. 2009). Only a few studies have used raw satellite signals (de Roos et al. 2024)."*

L.347: *"This means that FCOVER DA is more trivial than LAI DA, because the latter would require an additional function to connect CC to LAI, which could add uncertainty."*